# Endothelial cell-derived stem cell factor promotes lipid accumulation through c-Kit-mediated increase of lipogenic enzymes in brown adipocytes

Hyuek Jong Lee ®[1,5] ✉, Jueun Lee[2,5], Myung Jin Yang[1,3,5], Young-Chan Kim ®[1,3,5], Seon Pyo Hong[1], Jung Mo Kim[1], Geum-Sook Hwang ®[2,4] ✉ & Gou Young Koh ®[1,3] ✉

Active thermogenesis in the brown adipose tissue (BAT) facilitating the utilization of lipids and glucose is critical for maintaining body temperature and reducing metabolic diseases, whereas inactive BAT accumulates lipids in brown adipocytes (BAs), leading to BAT whitening. Although cellular crosstalk between endothelial cells (ECs) and adipocytes is essential for the transport and utilization of fatty acid in BAs, the angiocrine roles of ECs mediating this crosstalk remain poorly understood. Using single-nucleus RNA sequencing and knock-out male mice, we demonstrate that stem cell factor (SCF) derived from ECs upregulates gene expressions and protein levels of the enzymes for de novo lipogenesis, and promotes lipid accumulation by activating c-Kit in BAs. In the early phase of lipid accumulation induced by denervation or thermoneutrality, transiently expressed c-Kit on BAs increases the protein levels of the lipogenic enzymes via PI3K and AKT signaling. EC-specific *SCF* deletion and BA-specific *c-Kit* deletion attenuate the induction of the lipogenic enzymes and suppress the enlargement of lipid droplets in BAs after denervation or thermoneutrality in male mice. These data provide insight into SCF/c-Kit signaling as a regulator that promotes lipid accumulation through the increase of lipogenic enzymes in BAT when thermogenesis is inhibited.

As a specialized organ for non-shivering thermogenesis in response to sympathetic activation, brown adipose tissue (BAT) produces heat through uncoupling protein 1 (UCP1). Fatty acids (FAs) released from lipid droplets by lipolysis upon β3-adrenergic receptor (β3-AR) activation serve as a primary fuel for thermogenesis and stimulate UCP1 in brown adipocytes (BAs)[1,2]. Lipogenesis via de novo lipogenesis (DNL),

defined as a process of converting glucose to FAs, is required for thermogenesis in BAs, which is atypical because FA synthesis and FA oxidation antagonize each other in other tissues[3,4]. Therefore, the balance between lipolysis for FA supply and lipogenesis for FA storage is tightly regulated for the maintenance of homeostasis and thermogenesis in BAs. Activated BAT improves metabolic syndrome by

[1]Center for Vascular Research, Institute for Basic Science (IBS), Daejeon 34141, Republic of Korea. [2]Integrated Metabolomics Research Group, Western Seoul Center, Korea Basic Science Institute, Seoul 03760, Republic of Korea. [3]Graduate School of Medical Science and Engineering, Korea Advanced Institute of Science and Technology (KAIST), Daejeon 34141, Republic of Korea. [4]Colleage of Pharmacy, Chung-Ang University, Seoul 06974, Republic of Korea. [5]These authors contributed equally: Hyuek Jong Lee, Jueun Lee, Myung Jin Yang, Young-Chan Kim. ✉e-mail: hyuekjong.lee@gmail.com; gshwang@kbsi.re.kr; gykoh@kaist.ac.kr

accelerating FA and glucose usage[2,5]. In contrast, impaired thermogenic function that is observed in metabolic syndromes, aging, and thermoneutrality (the equilibrium point of heat production and loss) results in BAT whitening, characterized by reduced β-adrenergic signaling, mitochondrial dysfunction, and lipid accumulation[4,6–8]. Moreover, excessive lipid accumulation in BAs leads to BAT involution[9], characterized by decreased mitochondrial mass and increased lipid deposition[9]. Although the enzymatic machinery of DNL, including ATP-citrate lyase (ACL), acetyl-CoA carboxylase (ACC), fatty acid synthase (FASN), and stearoyl-CoA desaturase 1 (SCD1), is crucial for lipid accumulation in adipose tissues, their expression is decreased in BAT whitening induced by high-fat diet feeding, BAT denervation, or thermoneutrality[4,10,11]. Moreover, deletion of carbohydrate response element binding protein (ChREBP), a key transcription factor for DNL enzymes, inhibits FA synthesis in BAT at thermoneutrality, indicating that DNL is the central player for lipid accumulation in BAT whitening at thermoneutrality[9].

Endothelial cells (ECs) maintain organ homeostasis with distinct molecular and functional properties, and endothelial dysfunction initiates and progresses metabolic diseases[12–15]. The angiogenic response in adipose tissues is critical for maintaining normal function and proper remodeling of fat depots depending on energy consumption and metabolic demand[16–18]. Moreover, FA uptake and transport in ECs are crucial for lipid accumulation in white adipocytes (WAs)[19] because ECs regulate FA uptake in a paracrine manner[17,20–22]. In comparison, white adipocytes promote FA uptake and transport in ECs by regulating fatty acid transport proteins (FABPs)[23,24]. Therefore, the reciprocal interaction between ECs and white adipocytes is indispensable in lipid metabolism, insulin sensitivity, and adiposity in white adipose tissue (WAT)[25,26]. Although BAT, compared with WAT, is highly vascularized, the interaction between BAs and surrounding ECs in lipid metabolism has been poorly elucidated.

The receptor tyrosine kinase c-Kit and its only known ligand, stem cell factor (SCF), are expressed in various cells and tissues[27–30]. Activated c-Kit undertakes intracellular internalization and degradation and then activates phosphatidylinositol-3-kinase (PI3-K)/AKT, mitogen-activated protein kinase (MAPK)/ERK, Janus kinase/signal transducer, and activator of transcription (JAK/STAT) signaling depending on cell type[28,31]. SCF/c-Kit signaling is required for pancreatic development, glucose tolerance, and insulin secretion[31–33]. In addition, c-Kit expression is inversely associated with adiposity in both WAT and BAT, and blood SCF is increased in diet-induced obese mice, *db/db* mice, and human subjects with obesity[34], suggesting that c-Kit may play a role in lipid metabolism. However, the presence and roles of SCF/c-Kit signaling in BAT are unknown.

Here, we show that SCF/c-Kit signaling upregulates gene expression and protein level of the lipogenic enzymes and promotes lipid accumulation in BAs. SCF originates mainly in ECs, whereas c-Kit is present in classical BAs but not in WAs or beige cells. c-Kit⁺ BAs are transiently but dramatically increased in the early phase of lipid accumulation after denervation or thermoneutrality, but not by cold exposure. SCF/c-Kit signaling boosts lipogenic enzymes through the AKT signaling pathway, whereas SCF or c-Kit deficiency in ECs or BAs prevents the expansion of lipid droplets by attenuating the induction of lipogenic enzymes. Our results indicate that EC-derived SCF promotes lipid accumulation in BAs as a paracrine regulator for upregulating lipogenic enzymes in BAT.

## Results

### Potential crosstalk between ECs and BAs through SCF/c-Kit signaling

To identify an EC-specific role in BAT, we performed single-nucleus RNA sequencing (snRNA-seq) in the floating mixture of BAs and fragmented vasculature (FV) isolated from BAT (Fig. 1a). Through unsupervised clustering, we identified five distinct clusters- *Ucp1*high BA,

*UCP1*low BA, *Ucp1⁻* white adipocyte (WA), *Pecam1*high EC, and *PDGFR*βhigh pericyte (PC) clusters in the mixture (Fig. 1b, c), defined by differential expression of marker genes (Supplementary Fig. 1). The relative number of WAs was small but highly and selectively expressed *Slc7a10* and *Retn* (Fig. 1b–d), known WA-specific markers. Cell-to-cell interaction analysis based on ligand-receptor pairing using CellChat[35] distinguished enriched molecules involved in outgoing communication (signals from source cells) and incoming communication (signals to target cells) (Fig. 1e). Here, we noted four molecules (*Hspg, Kitl, Ephb,* and *Bmp*) in the ECs could interact with the cognate receptors in BAs through outgoing and incoming communications. Among these four molecules, *Kitl* receptor signaling pathways were highly enriched in the incoming communication of BAs (Fig. 1e). *c-Kit* expression was similarly high in both *Ucp1*high and *Ucp1*low BAs (Fig. 1f), which implies that the SCF highly secreted from the ECs may equally interact with the c-Kit of both BA clusters. Since the selective expressions of *SCF* and *c-Kit* between ECs *versus* BAs were distinctively higher than those of *Efnb1/Efnb2* and *Ephb2/Ephb4 or Bmp1/Bmp6* and *Bmpr1a/Bmpr2* (Fig. 1f), we chose and focused on investigating the roles of SCF/c-Kit signaling in the BAT.

Using *SCF*-GFP reporter mice[36], we verified the selective expression of SCF in the ECs of major metabolic organs. SCF was detected in more than 90% of CD31⁺ ECs and F4/80⁺ macrophages in BAT and WAT, while it was detected in 57% and 20% of CD31⁺ ECs in the pancreas and skeletal muscle, respectively (Supplementary Fig. 2a–e). For tracing c-Kit expression, we generated *c-Kit*iTR mice by crossing *c-Kit*-Cre-ER^T2 mice with tdTomato^flox/+ mice and then treated them with tamoxifen for 3 consecutive days to turn on c-Kit expression. Of note, c-Kit signal was detected in a certain population of BAs of BAT but was not detected at all in white adipocytes of WAT (Supplementary Fig. 3a). c-Kit was partially detected in CD31⁺ ECs in BAT, WATs, liver, and skeletal muscle but not in the pancreas (Supplementary Fig. 3a, b). However, abundant c-Kit⁺ cells were found in the non-EC population in the pancreas (Supplementary Fig. 3b), consistent with previous findings[31,37]. These results suggest that EC- or macrophage-derived SCF may communicate with c-Kit that is expressed almost exclusively in BAs.

### Thermogenic activation reduces c-Kit⁺ BA number and c-Kit protein level in BAT

Since BAT is a specialized organ for thermogenesis[38], we hypothesized that c-Kit in BAs might be involved in BAT thermogenesis. To test our hypothesis, we adapted *c-Kit*iTR mice to cold exposure (Cold, 7 °C) for 7 days to enhance thermogenesis and compared them with the mice exposed at room temperature (RT, 25 °C) (Supplementary Fig. 4a). The number of c-Kit⁺ BAs in BAT was 72.1% less in Cold versus RT, whereas the number of c-Kit⁺ ECs in inguinal white adipose tissue (iWAT) was comparable between the two groups (Supplementary Fig. 4b, c). Of note, although Cold generated beige cells that share the thermogenic function with BAs in WAT[39], c-Kit⁺ adipocytes were not detected in iWAT under either RT or Cold (Supplementary Fig. 4b). Because *c-Kit*iTR mice showed indelible and amplified labeling of *c-Kit*-expressing cells, we used immunoblotting to further analyze c-Kit protein in adipose tissues of C57BL/6 J mice adapted to Cold for 3 days or to RT or Cold for 7 days (Supplementary Fig. 4d). In BAT, UCP1 was 2.1-fold higher at day 3 in Cold than at day 7 at RT and maintained that level at day7 in Cold, whereas c-Kit was reduced in Cold at day 3 (75.3%) and day 7 (77.2%) (Supplementary Fig. 4e–g). Interestingly, the protein concentration of SCF in BAT was increased at day 3 (27.2 %) in Cold than RT and went back to RT level at day7 (Supplementary Fig. 4h), suggesting that increased SCF might activate c-Kit and internalized it in BAs[40,41]. Unlike BAT, the c-Kit in iWAT was unchanged despite UCP1 augmentation by beige cell recruitment in Cold[38] (Supplementary Fig. 4i–k).

To recapitulate induction of thermogenesis, we administered a β3-AR agonist (CL316,247) to *c-Kit*iTR mice for 5 consecutive days

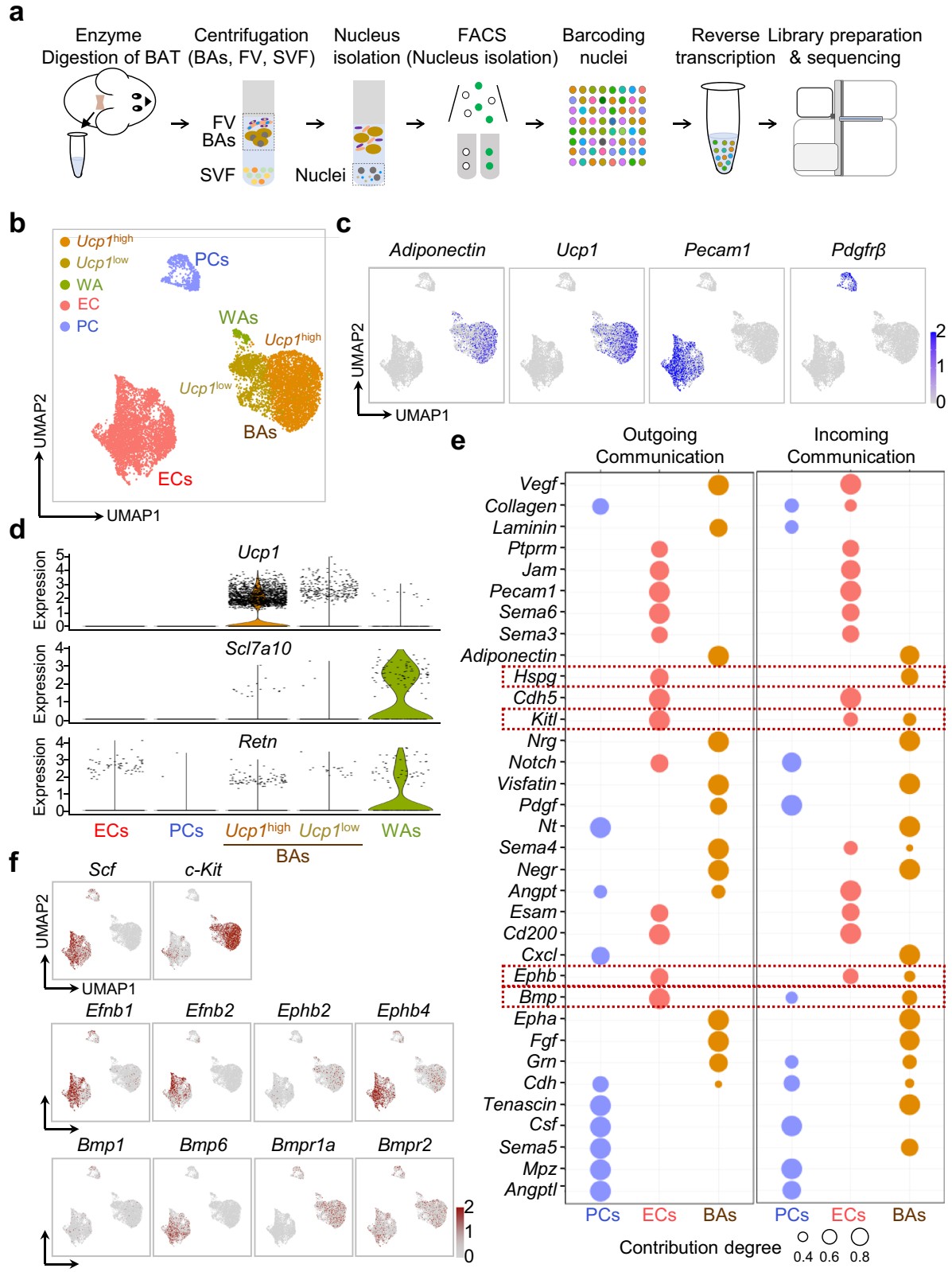

(Supplementary Fig. 5a). In agreement with the findings under Cold, compared with the mice receiving vehicle only as a control, the number of c-Kit⁺ BAs in BAT was 87.3% less with β3-AR agonist treatment, whereas the number of c-Kit⁺ ECs in iWAT was comparable between the vehicle- and agonist-treated mice (Supplementary Fig. 5b, c). However, c-Kit⁺ adipocytes were not detected in iWAT in either group (Supplementary Fig. 5b). In C57BL/6 J mice, BAT was isolated at day 3 in the

β3-AR agonist group and at day 5 in the vehicle and agonist-treated groups for immunoblotting (Supplementary Fig. 5d). β3-AR agonist treatment reduced c-Kit in BAT by 57.3% at day 3 and 38.6% at day 5, but UCP1 was 2.7-fold higher at day 3 and 2.4-fold higher at day 5 in agonist-treated mice compared with vehicle-treated mice at day 5 (Supplementary Fig. 5e–g). In contrast, c-Kit in iWAT was not significantly altered despite UCP1 augmentation by beige cell recruitment

**Fig. 1 | snRNA-seq analysis reveals distinct ligand-receptor pairings between ECs and BAs in BAT of adult mice. a** Diagram depicting the procedures for snRNA-seq on the isolated nuclei of BAs, ECs, and SCs of the BAT in adult C57BL/6 J mice (N = 20). snRNA-seq was performed in nuclei isolated from a floating layer consisting of brown adipocytes (BAs) and fragmented vasculature (FV) after BAT digestion. **b** UMAP plot showing 5 main clusters, *Ucp1*high and *Ucp1*low BAs, white adipocytes (WAs), endothelial cells (ECs), and pericytes (PCs) of a total of 10,841 cells in BATs. **c** UMAP plots showing the expression of the distinct genes in each cluster of BAs, ECs, and PCs. **d** Violin plots showing the expressions of indicated genes in *Ucp1*high and *Ucp1*low BAs and WAs. **e** Dot plots showing indicated signaling pathways from source cells (outgoing communication) to target cells (incoming communication) according to ligand-receptor pairings among BAs, ECs, and PCs. The signaling pathways from ECs to BAs are highlighted by dark red dotted-lined boxes. Relative contribution degrees are present as different sizes of circles. **f** UMAP plot showing ligand-receptor pairing for *Kit*, *Efnb*, and *Bmp* signaling pathways in 5 main clusters.

even after β3-AR agonist treatment[42] (Supplementary Fig. 5h–j). Collectively, these findings imply that the presence of c-Kit is limited only to classical BAs, not to beige cells, and c-Kit⁺ BAs and c-Kit protein are reduced by activating thermogenesis.

### Sympathetic denervation increases the number of c-Kit⁺ BAs and c-Kit protein level in BAT

Sympathetic denervation of BAT results in UCP1 downregulation, thermogenesis impairment, and abnormal lipid storage in BAs by inhibiting β3-AR signaling, eventually leading to BAT whitening[10,43,44]. To elucidate whether c-Kit is altered upon deactivation of thermogenesis in BAT, we employed a BAT unilateral-denervation (UDN) model in *c-Kit*iTR mice[10,45]. Left intercostal nerves on BATs were cut and removed (D-BAT), while the right nerves were left intact (I-BAT) as an internal control (Fig. 2a). Both I-BAT and D-BAT were separately isolated at days 3 and 5 after UDN with tamoxifen treatment (Fig. 2b). We confirmed the denervation in the D-BAT by measuring the protein level of tyrosine hydroxylase (TH), a specific enzyme for sympathetic neurons. The TH levels in D-BAT were 60-70% less than those in I-BAT on day 3 after UDN (Fig. 2c, d). Intriguingly, the number of c-Kit⁺ BAs was 15.8-fold and 10.1-fold higher in D-BAT compared with I-BAT at days 3 and 5 (Fig. 2e, f).

We then analyzed lipid accumulation in BAs after UDN by measuring the size of lipid droplets. Droplet size was defined as an area of BODIPY⁺ lipid droplet and categorized into four size-based subpopulations, as follows: small-sized, 0–10 μm²; medium-sized, 12–20 μm²; large-sized, 21–50 μm²; and extra-large-sized, >51 μm². Consistent with previous reports[43,44], most lipid droplets (79.9%) in I-BAT at day 3 and 5 were in small-sized group, whereas D-BAT at day 3 and 5 had more large-sized lipid droplets and fewer small-sized lipid droplets (Fig. 2e, g), implying enlargement of lipid droplets in D-BAT at day 3 and 5. In line with immunofluorescence staining results, c-Kit protein was 1.8-fold higher at day 3 in D-BAT compared with I-BAT, but not at day 5 (Fig. 2h, i). The latter could be due to a reduced *c-Kit* expression by rapid metabolic adaptation, an enhanced ligand-induced internalization and degradation of c-Kit, or both, since SCF concentration in BAT was 3.31-fold increased in D-BAT at day 4 after UDN (Fig. 2k), indicating that SCF was a primary cause for the reduction of c-Kit[40,41] in BAT at day 5 after UDN. UCP1 proteins were 47% reduced at day 3 and 51% reduced at day 5 in D-BAT compared with I-BAT (Fig. 2h, j). Together, these findings indicate that UDN transiently increased the number of c-Kit⁺ BAs along with UCP1 reduction and lipid droplet enlargement in BAs.

### c-Kit activation induces DNL enzymes in cultured BAs

To uncover the roles of c-Kit in BAs, we performed snRNA-seq in the floating layer of D-BAT at day 3 after denervation (Supplementary Fig. 6a) and focused on analyzing only the BAs cluster among the three clusters obtained (Supplementary Fig. 6b). *c-Kit*⁺ BAs and *c-Kit*⁻ BAs constituted 15.6% and 84.3% of the total BAs of the D-BAT3 dataset (Supplementary Fig. 6b). We compared differentially expressed genes between *c-Kit*⁺ BAs *versus* *c-Kit*⁻ BAs using the volcano plotting (Supplementary Fig. 6c). Of interest, expressions of lipogenesis-related genes (*Fasn*, *Scd1*, *Acly*, *Thrsp*, and *Slc25a1*) and a glycogen metabolism-related gene (*Ppp1r3b*) were higher in *c-Kit*⁺ BAs compared with *c-Kit*⁻ BAs (Supplementary Fig. 6c). These results show that

lipogenesis-related genes, more specifically DNL enzymes (*Acly* and *Scd*), are upregulated in *c-Kit*⁺ BAs in D-BAT3.

To validate the upregulation of DNL enzyme genes in *c-Kit*⁺ BAs, we investigated whether SCF-induced c-Kit activation enhances DNL enzymes in the cultured BAs (see Method).

To validate whether the cultured BAs differentiate and considering the experimental setting, the BAs were transfected with c-Kit cDNA inserted (c-Kit overexpressed) or empty lentiviral plasmid (Control) tagged with GFP encoding gene, and were differentiated under differentiation media and induction media (Supplementary Fig. 7a). Both c-Kit–overexpressed and Control cultured BAs were similarly differentiated by gaining lipid droplets and upregulated expression of *Ucp1* mRNA (Supplementary Fig. 7b, c). After 12 h of starvation with 1% fetal bovine serum (FBS), either bovine serum albumin (BSA, 100 ng/mL) or SCF (100 ng/mL) was treated into c-Kit–overexpressed BAs for 12 h. Compared with BSA treatment, SCF treatment reduced c-Kit by 58.4% at 6 h and 78.2% at 12 h, as previously described[40,41]. At 10 min after SCF treatment, the phosphorylations of AKT and ERK, representative downstream molecules of the c-Kit pathway, were increased [pAKT1 (Ser473) by 9.6-fold, pAKT2 (Ser474) by 12.4-fold, and pERK1/2 (Thr202/Tyr204) by 12.3-fold] (Fig. 3a, b). Additionally, SCF treatment increased protein levels of ACL (2.1-fold) and ACC (1.5-fold) at 12 h (Fig. 3a, b). Surprisingly, SCD1 was dramatically increased by SCF treatment (4.1-fold at 6 h and 6.3-fold at 12 h) (Fig. 3a, b). In comparison, either BSA or SCF treatment did not alter AKT and ERK1/2 phosphorylations and protein levels of ACL, ACC, and SCD1 in Control (transfected with empty lentiviral plasmid) BAs for 12 h (Fig. 3a, b). Nevertheless, the protein level of FASN was not altered either by BSA or SCF treatment in Control or c-Kit–overexpressed BAs (Fig. 3a, b). Thus, the activations of intracellular signaling Akt and ERK and upregulation of DLN enzymes are specifically mediated through SCF/c-Kit signaling.

To examine whether activation of c-Kit signaling actually enhances lipid accumulation, SCF (100 ng/ml) or BSA (100 ng/ml) was added into the c-Kit–overexpressed preadipocytes for 7 days, which were grown in the induction media without insulin supplement (Supplementary Fig. 7d). Compared with BSA treatment, SCF treatment showed 1.64-fold higher LipidTOX⁺ lipid area in the cultured BAs (Supplementary Fig. 7e, f), implying that activation of c-Kit signaling enhances lipid accumulation in BAs mediated through upregulation of DNL enzymes.

The key transcriptional factors for DNL enzymes are sterol regulatory element-binding protein 1 (SREBP1)[46–49] and ChREBP[50,51], and of importance, AKT is necessary for stimulating both transcriptional factors[4,52]. To examine whether c-Kit regulates SCD1 via AKT, several inhibitors targeting c-Kit downstream signaling pathways at different levels were pretreated for 1 h before BSA, VEGF, or SCF treatment. Inhibitors such as dynasore (20 μM, a cell-permeable inhibitor of clathrin-dependent internalization), SU5402 (1 μM, a potent multi-targeted receptor tyrosine kinase inhibitor), wortmannin (30 nM, a specific PI3K inhibitor), or AKT1/2 inhibitor (25 nM, a potent isozyme selective AKT1/2 kinase inhibitor) almost completely inhibited induction of SCD1 at 12 h after SCF treatment (Fig. 3c, d). However, inhibitors including PD0325901 (1 μM, a selective and non-ATP-competitive MEK inhibitor), ruxolitinib (10 μM, a potent and selective JAK1 and JAK2 inhibitor), and Y27632 (10 μM, a selective ROCK inhibitor) could not block the increase in SCD1 after SCF treatment (Fig. 3c, d). These

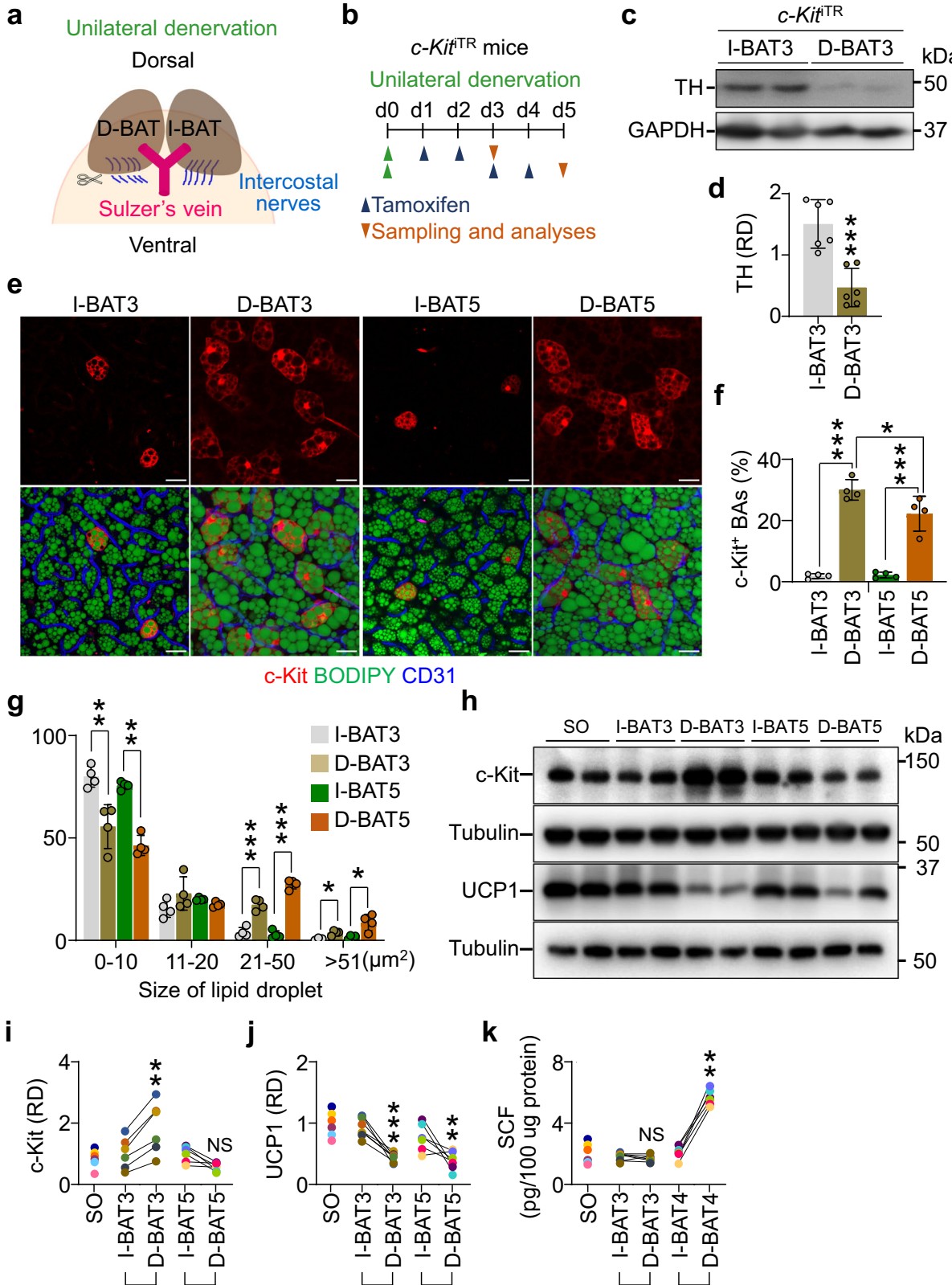

results imply that SCF/c-Kit signaling positively regulates protein levels of DNL enzymes via the PI3K and AKT intracellular signaling pathways in BAs.

## c-Kit deletion in BAs reduces DNL enzymes after denervation

To further clarify the function of c-Kit in protein levels of DNL enzymes, we generated BA-specific *c-Kit* deletion (*c-Kit*<sup>ΔBA</sup>) mice by crossing *UCP1*-Cre mice with *c-Kit*<sup>flox/flox</sup> mice[53] (Supplementary Fig. 8a). In 8-week-old *c-Kit*<sup>ΔBA</sup> mice compared with control mice, c-Kit was decreased only in BAT (77.8%) and not in WATs or pancreas. At baseline, body weight, fat weight, protein levels of DNL enzymes, BA lipid droplet size, and BA size were comparable between the groups (Supplementary Fig. 8b–i). Moreover, differences in mitochondrial sizes and areas and protein levels of PGC1-α and UCP1 in the BAs between

**Fig. 2 | Denervation increases the number of c-Kit⁺ BAs and protein level of c-Kit in BAT. a, b** Diagram depicting the experimental procedure for the unilateral denervation (D-BAT) and contralateral sham-operation (intact BAT, I-BAT) in the interscapular BAT of 8-week-old *c-Kit*ⁱᵀᴿ mice, tamoxifen administrations, and sampling 3 or 5 days later for the analyses. **c, d** Representative immunoblotting and comparisons of tyrosine hydroxylase (TH) in I-BAT and D-BAT at day 3 after unilateral denervation (I-BAT3 or D-BAT3). The same amount of protein loading in each lane is verified by immunoblotting of GAPDH. Each dot indicates a value from one mouse and *n* = 6 mice/group from two independent experiments. Vertical bars indicate mean ± SD. ***P < 0.001 versus I-BAT3 by two-tailed t-test. Protein sizes are indicated as kilodalton (kDa). **e–g,** Representative images and comparisons of populations of different sizes of lipid droplets in I-BAT and D-BAT and the number of c-Kit⁺ BAs in BAT from *c-Kit*ⁱᵀᴿ mice at day 3 or 5 after unilateral denervation (I-BAT3, D-BAT3, I-BAT5, or D-BAT5). Scale bars, 20 μm. Each dot indicates a value from one mouse and *n* = 4 mice/group from two independent experiments. Vertical bars indicate mean ± SD. ***P < 0.001 versus I-BAT and *P < 0.05 versus D-BAT3 by

one-way ANOVA test followed by Tukey's *post-hoc* test (**d**), *P < 0.05, **P < 0.01 and ***P < 0.001 versus I-BAT by two-tailed t-test (**e**). **h–j** Unilateral denervation in the interscapular BAT of 8-week-old C57BL/6 J mice was performed and BATs were isolated 3 or 5 days later for the analyses (I-BAT3, D-BAT3, I-BAT5, or D-BAT5). Representative immunoblotting and comparisons of relative density (RD) of c-Kit and UCP1 in BAT of sham-operated (SO), I-BAT, and D-BAT. The same amount of protein loading in each lane is verified by immunoblotting of tubulin. Each dot indicates a value from one mouse and *n* = 6 mice/group from two independent experiments. Vertical bars indicate mean ± SD. **P < 0.01 and ***P < 0.001 versus I-BAT3 or I-BAT5 by one-way ANOVA test followed by Tukey's *post-hoc* test. NS, not significant. Protein sizes are indicated as kilodalton (kDa). **k** Comparisons of SCF concentration in the BAT of sham-operated (SO), I-BAT, and D-BAT at day 3 or 4 after unilateral denervation (I-BAT3, D-BAT3, I-BAT4, or D-BAT4). Each dot indicates a value from one mouse and n = 6 mice/group from two independent experiments. Vertical bars indicate mean ± SD. **P < 0.01 versus I-BAT3 or I-BAT4 by one-way ANOVA test followed by Tukey's *post-hoc* test. NS not significant.

control and *c-Kit*ᐞᴮᴬ mice were not detected (Supplementary Fig. 9a–e). To address whether c-Kit signaling truly plays a functional role in BAT thermogenesis, we adapted *c-Kit*ᐞᴮᴬ mice to RT or Cold for 5 days and conducted calorimetry measurements (Supplementary Fig. 9f). Unexpectedly, BA-specific *c-Kit* deletion did not lead to any differences in UCP1 protein level or metabolic rate as measured by oxygen and carbon dioxide consumption in RT or Cold (Supplementary Fig. 9g–o), suggesting that c-Kit does not alter thermogenesis in BAT, which is inconsistent with the findings of Hwang et al.[34]. using the global *c-Kit*ᵐᵘᵗᵃⁿᵗ mice.

Next, we performed UDN in *c-Kit*ᐞᴮᴬ mice to interrogate the effect of c-Kit on protein levels of DNL enzymes by conducting protein analysis at day 3 and morphologic analysis at day 5 in the separately isolated I-BAT and D-BAT mice (Fig. 4a). In control mice, ACC and FASN were reduced by 47.9% and 50% in D-BAT3 compared with I-BAT3 (Fig. 4b, c), consistent with previous results[10]. However, in *c-Kit*ᐞᴮᴬ mice, only SCD1 was markedly reduced by 63.3% in D-BAT compared with I-BAT (Fig. 4b, c). As a central regulator of DNL, SCD1 modulates FA accumulation and the FA component of cellular lipids by converting palmitate (C16:0) into palmitoleate (C16:1) and stearate (C18:0) into oleate (C18:1)[54,55]. Therefore, we postulated that the BA-specific *c-Kit* deletion might affect lipid droplet size in BAs. To address this idea, we measured and compared the sizes of lipid droplets in I-BAT and D-BAT from control and *c-Kit*ⁱᵀᴿ mice. In both groups, UDN generated large-sized lipid droplets, and population of small-sized lipid droplets was diminished. However, to our surprise, BA-specific *c-Kit* deletion prevented the production of extra-large-sized lipid droplets after UDN (Fig. 4d, e). Collectively, these results indicate that c-Kit contributes to the enlargement of lipid droplets by preventing the reduction of SCD1 in BAs after UDN.

### c-Kit deletion in BAs reduces lipogenic enzymes at thermoneutrality

To validate and expand our observations on the UDN mouse model, we adapted the mice to a thermoneutrality condition that naturally leads to BAT whitening with lipid deposition and mitochondrial mass reduction. To delineate whether c-Kit is altered at thermoneutrality, we adapted *c-Kit*ⁱᵀᴿ mice to RT (25 °C) or thermoneutrality (30 °C) for 3 or 5 days (TN3, TN5) (Fig. 5a). In line with the findings of the UDN, thermoneutrality increased the number of c-Kit⁺ BAs at day 3 (17.4-fold) or at day 5 (12.4-fold) (Fig. 5b, c). In addition, thermoneutrality increased the size of lipid droplets and reduced the population of small-sized lipid droplets at day 3 and day 5 (Fig. 5b, d). We additionally analyzed the expression of c-Kit and DNL enzymes in BAT from C57BL/6 J mice at day 3 and day 5 after adaptation at thermoneutrality (Fig. 5e). c-Kit was 2.2-fold higher only at day 3 and then returned to RT level at day 5. ACL was decreased by 22.4% and SCD1 by 65.0% on day 5, but ACC and FASN were unchanged (Fig. 5f–k). These results indicate that

thermoneutrality transiently increased c-Kit and subsequently decreased ACL and SCD1 despite lipid accumulation in BAs.

Next, we questioned whether BA-specific *c-Kit* deletion could diminish DNL enzyme levels and lipid accumulation under thermoneutrality. To address this question, we adapted *c-Kit*ᐞᴮᴬ mice to thermoneutrality for 3 or 5 days (TN3 or TN5) (Fig. 6a) and found similar degrees of substantial decreases in ACL (35.8% and 50.9%), ACC (31.1% and 40.6%), FASN (55.1% and 48.6%), and SCD1 (61.1% and 65.5%) at TN3 and TN5 in BAT of *c-Kit*ᐞᴮᴬ mice (Fig. 6b–f). Since there were no significant differences in the reductions of the enzymes between TN3 and TN5, the lack of c-Kit does not seem to generate a fast or delayed adaptative response to the TN. To elucidate whether SCD1 activity changes under c-Kit signaling, we performed ultra-performance liquid chromatography coupled with a hybrid quadrupole orthogonal time-of-flight mass spectrometer (UPLC − QTOF − MS) analysis in BAT at day 3 after thermoneutrality and measured key free FAs (FFAs) related to SCD1 activity. In BATs from *c-Kit*ᐞᴮᴬ mice compared with controls, the proportion of palmitoleate (C16:1) in the total FFA fraction was lower by 24.5% and the SCD1 desaturation index (C16:1 to C16:0 ratio) was 18.7% reduced. However, the proportions of stearate (C18:0) and oleate (C18:1) in total FFAs did not differ between the two groups (Fig. 6g). Our findings imply that palmitoleate (C16:1) rather than oleate (C18:1) is presumably a dominant product by c-Kit-mediated increased SCD1 activity in BAT. Accordingly, the population of medium- and large-sized lipid droplets was increased, but those in the extra-large group were dramatically decreased in *c-Kit*ᐞᴮᴬ mice at day 5 of thermoneutrality (Fig. 6h, i). Together, these findings lead us to conclude that c-Kit induces lipid accumulation by upregulation of DNL enzymes, triggering the enlargement of lipid droplets in BAs at thermoneutrality.

### EC-derived SCF promotes lipid accumulation via c-Kit signaling in BAs at thermoneutrality

Finally, we asked whether EC-derived SCF could positively regulate c-Kit-mediated lipid accumulation in BAs. Our previous results (Supplementary Fig. 2a–e) had shown that SCF was produced by ECs and macrophages in adipose tissues. Thus, we examined SCF expression in BAT using EC-specific *SCF* deletion mice (*SCF*ⁱᐞᴱᶜ) generated by crossing *VE-cadherin*-Cre-ERᵀ² mice and *SCF*ᶠˡᵒˣ/ᶠˡᵒˣ mice (Supplementary Fig. 10a). For determining the deletion efficiency, adipose tissues in *SCF*ⁱᐞᴱᶜ mice were obtained at day 7 after the first tamoxifen treatment (Supplementary Fig. 10a). Compared with control mice, the concentration of SCF in *SCF*ⁱᐞᴱᶜ mice, as measured by ELISA, was significantly reduced in BAT (94%), iWAT (83%), epididymal WAT (76%), and plasma (63%) (Supplementary Fig. 10b). However, SCF deletion in ECs did not yield any significant change in body or fat weight or in levels of c-Kit and lipogenic enzymes in BAT at baseline (Supplementary Fig. 10c–i). Because SCF/c-Kit signaling plays an essential role in insulin secretion from pancreatic β-cells[32,33], we measured fasting blood glucose weekly

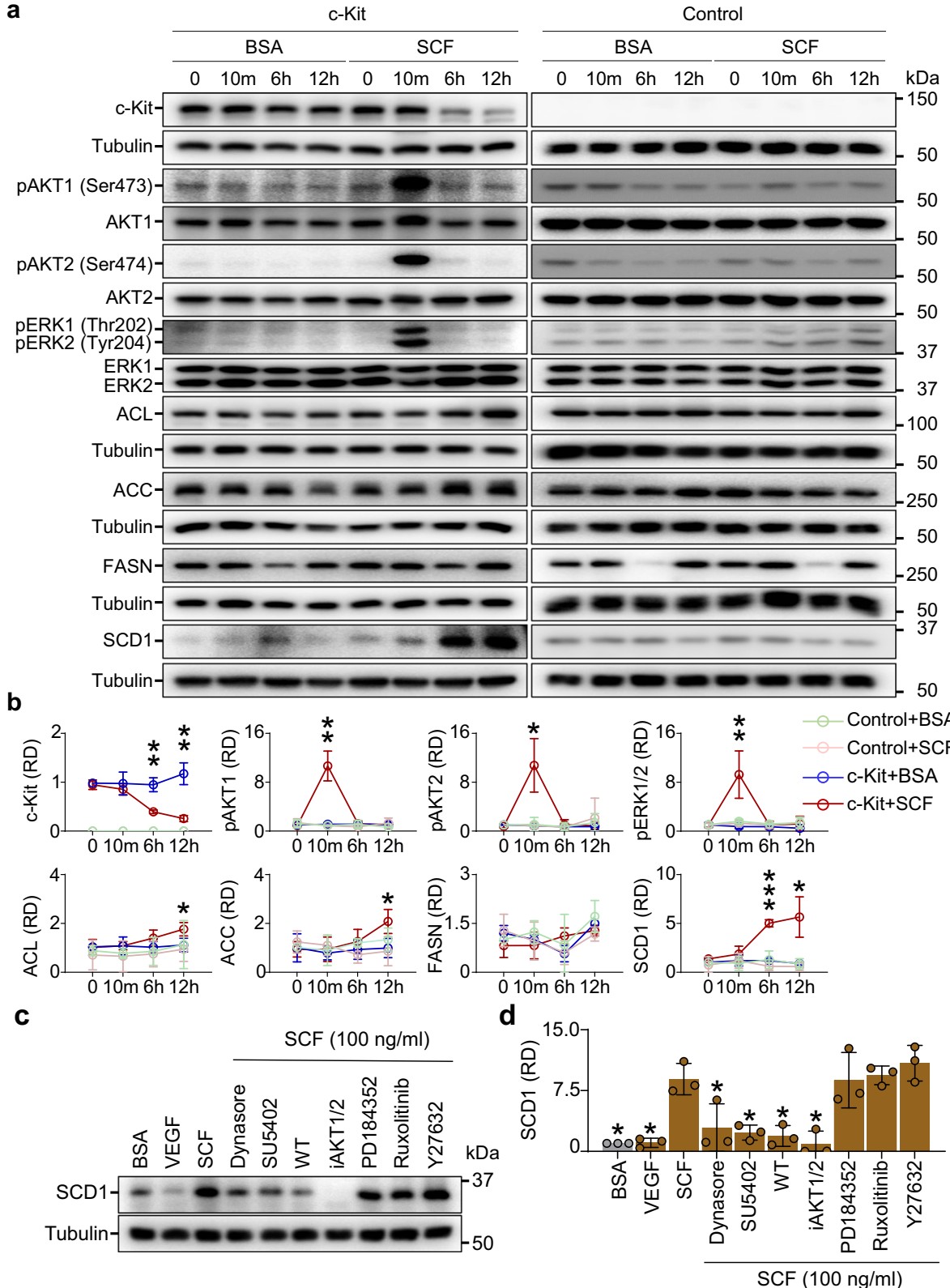

in *SCF*[iΔEC] mice during 8–11 weeks of age. For at least the first 2–3 weeks after SCF was turned off in the ECs by tamoxifen, fasting blood glucose levels were unaffected in *SCF*[iΔEC] mice (Supplementary Fig. 10j), indicating that pancreatic insulin secretion was unimpaired.

To characterize the expression of SCF in macrophages, we generated macrophage-specific *SCF* deletion mice (*SCF*[ΔMΦ]) by crossing *LysM*-Cre mice and *SCF*[flox/flox] mice (Supplementary Fig. 11a). In contrast

to *SCF*[iΔEC] mice, the concentrations of SCF in adipose tissues and plasma were unchanged in *SCF*[ΔMΦ] mice (Supplementary Fig. 11b–d). These results imply that SCF in BAT is almost exclusively expressed in ECs than in macrophages.

To determine whether EC-derived SCF activates *c*-Kit-mediated lipid accumulation in BAs, *SCF*[iΔEC] mice were adapted to thermoneutrality for 3 or 5 days (Fig. 7a). c-Kit protein was increased 3.7- and

**Fig. 3 | SCF increases DNL enzymes by activating c-Kit signaling pathway in cultured BAs. a, b** Representative immunoblotting and comparisons of c-Kit, pAKT1 (Ser473), AKT1, pAKT2 (Ser474), AKT2, pERK1/2 (Thr202/Tyr204), ERK1/2, ACL, ACC, FASN, and SCD1 before and 10 min (10 m), 6 h, and 12 h after either bovine serum albumin (BSA, 100 ng/ml) or SCF (100 ng/ml) treatment in Control (transfected with empty lentiviral plasmid) or c-Kit over-expressed (transfected with c-Kit cDNA inserted lentiviral plasmid) cultured BAs. The cultured BAs were pre-incubated in high glucose DMEM containing 1% FBS for 12 h before and after the treatment. The same amount of protein loading in each lane is verified by immunoblotting of tubulin. Dots and bars indicate mean ± SD from $n = 3$/group from two independent experiments. *$P < 0.05$, **$P < 0.01$, and ***$P < 0.001$ versus BSA by two-tailed t-test. Protein sizes are indicated as kilodalton (kDa). **c, d** Representative

immunoblotting and comparison of effect of indicated signaling inhibitor on the SCF-induced increased protein level of SCD1 in the c-Kit over-expressed and differentiated cultured BAs. The cultured BAs were pre-incubated in high glucose DMEM containing 1% FBS for 12 h before and after the treatment. For the inhibitor experiments, either Dynasore (20 μM), SU5402 (1 μM), wortmanin (30 nM), AKT1/2 inhibitor (25 nM), PD 0325901 (1 μM), ruxolitinib (10 μm) or Y27632 (10 mM) was treated to cultured cells for 1 hr, and then SCF (100 ng/ml) or VEGF (100 ng/ml) was added and incubated for 12 h. Each dot indicates a value from one experiment and $n = 3$/group from two independent experiments. Vertical bars indicate mean ± SD. *$P < 0.05$ versus SCF by one-way ANOVA test followed by Tukey's post-hoc test. Protein sizes are indicated as kilodalton (kDa).

3.6-fold at day 3 and 5 of thermoneutrality in BAT of $SCF^{i\Delta EC}$ mice, while it was increased only 2.3- and 1.6-fold at day 3 and 5 of thermoneutrality in BAT of control mice (Fig. 7b, c). These data imply that the SCF deletion from the adjacent ECs diminishes the ligand-dependent internalization and degradation of c-Kit[40,41], which leads to higher levels of c-Kit in the cell membrane of c-Kit+ BAs in the $SCF^{i\Delta EC}$ mice compared with control mice. In addition, compared with BAT from control mice, ACC (79.4%) and SCD1 (79.6%) were decreased in BAT of $SCF^{i\Delta EC}$ mice at TN3, and ACC (58.8%), FASN (47.3%), and SCD1 (83.9%) were decreased in BAT of $SCF^{i\Delta EC}$ mice at TN5 (Fig. 7d–h). To determine whether SCD1 activity in BAT at day 3 of thermoneutrality is altered by SCF ablation in ECs, we measured key FFAs related to SCD1 activity using UPLC – QTOF – MS analysis. The proportion of palmitoleate (C16:1) in total FFAs declined by 45.7% and the SCD1 desaturation index by 39.6% in BAT from $SCF^{i\Delta EC}$ compared with control mice. However, the proportions of stearate (C18:0) and oleate (C18:1) did not differ between the two groups (Fig. 7i), similar to the findings in $c$-$Kit^{\Delta BA}$ mice. Additionally, the population of large-sized lipid droplets increased but the number of extra-large-sized lipid droplets dramatically declined in $SCF^{i\Delta EC}$ mice (Fig. 7j, k). These data suggest that EC-derived SCF enhances lipid accumulation via c-Kit–mediated induction of lipogenic enzymes and promotes the growth of lipid droplets in BAs.

## Discussion

In this study, we demonstrated that SCF is mainly localized in vascular ECs, interacts with c-Kit expressing BAs, and promotes lipid accumulation in BAs by positively regulating DNL enzymes. c-Kit+ BAs transiently increased only in BAT after UDN or thermoneutrality, which induces lipid accumulation and mitochondrial inactivation. As a consequence, SCF or c-Kit deficiency in ECs or BAs, respectively, prevents the expansion of lipid droplets by reducing DNL enzymes (Fig. 8).

SCF/c-Kit signaling is crucial for the growth and function of multiple lineages of progenitor cells and pancreatic β-cells, and c-Kit may be involved in peripheral lipid metabolism[32,34]. Impairment of β-cell growth and function in global c-Kit mutant mice compromises insulin secretion associated with various physiological processes[32,33], which from a metabolic perspective has distracted attention from the roles of SCF and c-Kit in organs other than the pancreas. Our analysis with snRNA-seq and immunofluorescence staining using reporter mice clearly showed the presence of SCF in ECs and c-Kit in BAs of BAT. Interestingly, c-Kit was detected only in a certain population of classical BAs and not in white adipocytes, beige cells, hepatocytes, or myocytes at the basal level, implying that c-Kit plays a selective role in BAs only. Indeed, our findings clearly show that c-Kit plays a promoting role in lipid accumulation rather than lipolysis in the BAs. Moreover, we found that BA-specific c-Kit ablation does not affect thermogenesis induced by cold exposure. These findings are inconsistent with those of Hwang et al.[34]. using the global $c$-$Kit^{mutant}$ mice, which show reduced thermogenesis. The main reason for this difference could be attributed to the different metabolic phenotypes between the $c$-$Kit^{\Delta BA}$ mice and the $c$-$Kit^{mutant}$ mice. The $c$-$Kit^{\Delta BA}$ mice used in this study were 8–10 weeks old and exhibited no apparent alterations in the metabolic organs. In fact,

the BATs of $c$-$Kit^{\Delta BA}$ mice have no alterations in total size, lipid droplet size, protein levels of DNL enzymes, PGC1-α and UCP1, and mitochondrial sizes and areas, which leads to no alteration in thermogenesis induced by cold exposure. In contrast, the $c$-$Kit^{mutant}$ mice used and analyzed in their study[34] were 20–40 weeks old and exhibited multiple defects in the major metabolic organs. In particular, the BAT and skeletal muscle, major thermogenic organs, of 20-weeks-old $c$-$Kit^{mutant}$ mice have reduced number, mtDNA, and respiration in mitochondria and reduced PGC1-α protein level[34], which leads to reduced thermogenesis. We attribute these alterations of BAT to impairment of insulin secretion from the compromised pancreas or metabolic alterations of other major metabolic organs in the $c$-$Kit^{mutant}$ mice[32,33], since our $c$-$Kit^{\Delta BA}$ mice did not show any alterations in the BAT and thermogenesis.

We also showed that lipid droplets grow rapidly by inhibiting thermogenesis or sympathetic activation. However, the underlying mechanism of lipid accumulation in these conditions has not been fully characterized. Using snRNA-seq and knock-out mice, we found that c-Kit in BAs is connected with lipogenesis. DNL initiates a series of enzymatic reactions for converting glucose into FAs that are later esterified for triglyceride synthesis. Both SREBP1-c and ChREBP transcriptionally regulate DNL enzymes, and their activity is determined by AKT[4,52]. DNL enzymes are decreased under abnormal lipid accumulation in white adipocytes[56], which may lead to insulin resistance based on induction of WAT inflammation and systemic insulin resistance in adipocyte-specific ChREBP deletion mice[57]. Hyperleptinemia is a possible mechanism for reducing DNL enzymes in obesity-related WAT[58], but whether leptin plays the same role in BAs after UDN or thermoneutrality is unclear. Our data indicate that c-Kit+ BAs are dramatically and transiently increased in BAT after UDN or thermoneutrality and that c-Kit activation enhances the augmentation of DNL enzymes through AKT activation in cultured BAs. Conversely, ablation of SCF or c-Kit in BAT reduces DNL enzyme levels after UDN or thermoneutrality, emphasizing that c-Kit prevents the reduction of DNL enzymes in lipid accumulation. As a rate-limiting enzyme, SCD1 modulates FA accumulation and the FA component of cellular lipids by converting palmitate (C16:0) into palmitoleate (C16:1) and stearate (C18:0) into oleate (C18:1)[54,55]. In our observations, c-Kit activation by SCF in vitro or ablation of SCF or c-Kit in BAT preferentially manipulates SCD1 more than other DNL enzymes, so that reduction of SCD1 and its activity by SCF or c-Kit deficiency inhibits expansion of lipid droplets in BAs after UDN and thermoneutrality. In addition, our findings indicate that the SCF/c-Kit signaling-induced SCD1 seems to facilitate the conversion of palmitate to palmitoleate rather than the conversion of stearate to oleate in the BAT. In fact, the preferential conversion of palmitate to palmitoleate or stearate to oleate by SCD1 is fat-depot specific[59–61]. Therefore, it is warranted to investigate the preferential conversion of SCD1 in the BAT under several different conditions as a future study.

In WAT, the reciprocal interaction between ECs and white adipocytes is crucial for lipid accumulation. Although vascular refraction contributes to BAT whitening by hypoxia-induced β-AR inhibition[8], it is unclear whether the vasculature is directly related to lipogenesis in BAT.

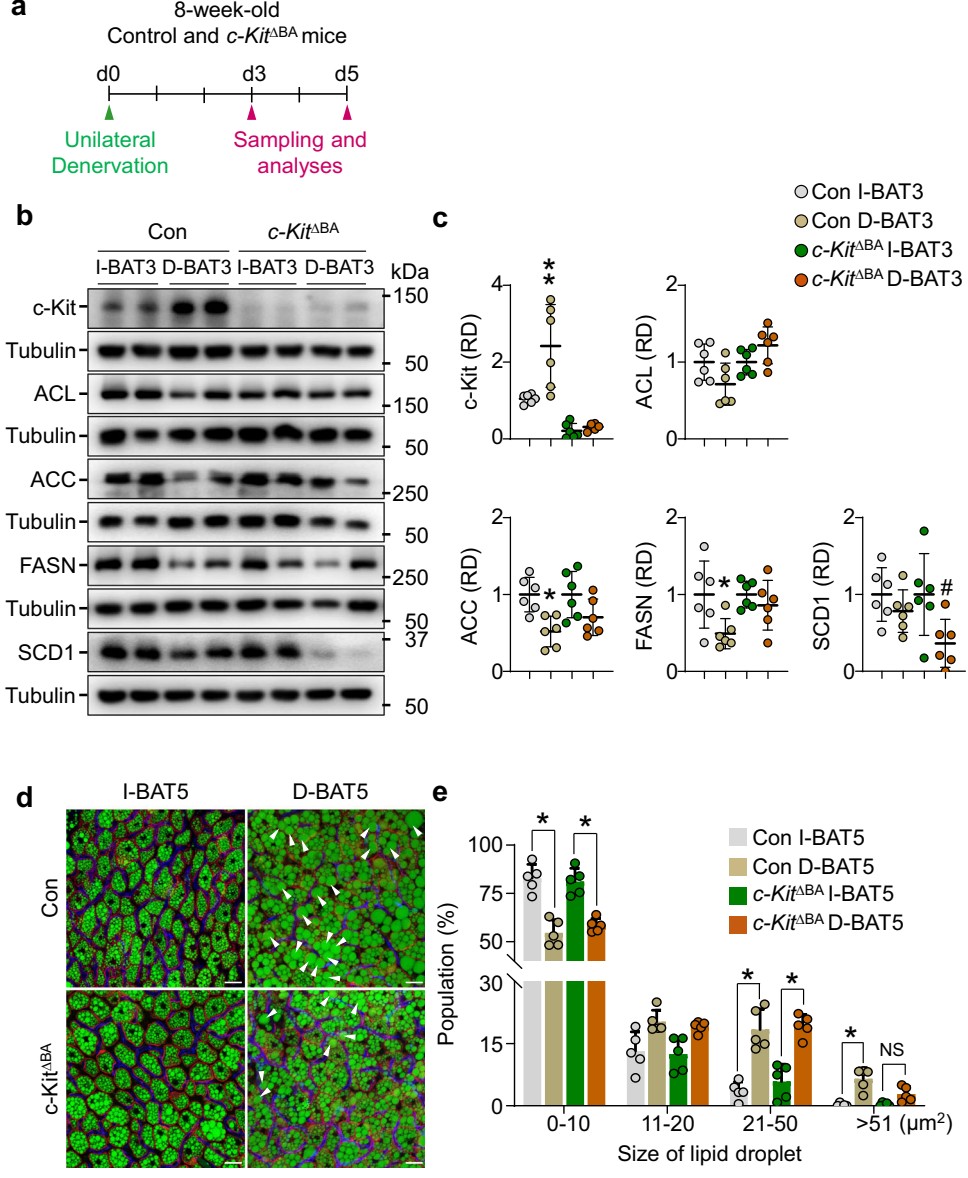

**Fig. 4 | BA-specific deletion of c-Kit inhibits the denervation-induced lipid accumulation and reduces SCD1 in the BAs of BAT. a** Diagram of the experimental procedure for the unilateral denervation of BAT in 8-week-old control (Con) and *c-Kit*ΔBA mice, and harvest BAT (I-BAT and D-BAT) 3 or 5 days later for the analyses. **b, c** Representative immunoblotting and comparisons of c-Kit, ACL, ACC, FASN, and SCD1 in the I-BAT and D-BAT 3 days after the unilateral denervation (I-BAT3 or D-BAT3). The same amount of protein loading in each lane is verified by immunoblotting of tubulin. Each dot indicates a value from one mouse and *n* = 6 mice/group from two independent experiments. Horizontal and vertical bars indicate mean ± SD. *,#P < 0.05 versus Con I-BAT3 or *c-Kit*ΔBA I-BAT3 by one-way ANOVA test followed by Tukey's post-hoc test. Protein sizes are indicated as kilodalton (kDa). **d, e** Representative images and comparisons of the populations of different sizes of lipid droplets in the CD147+ BAs of BAT among the indicated groups. White arrowheads indicate the lipid droplets whose size is larger than 51 μm² in CD147+ BA. Scale bars, 20 μm. Each dot indicates the population (%) of the total of 1200–2400 droplets (100%) from three portions of BAT in one mouse and *n* = 5 mice/group from two independent experiments. Vertical bars indicate mean ± SD. *P < 0.05 by two-way ANOVA test followed by Sidak test. NS not significant.

Our results emphasize that SCF from ECs is a paracrine regulator for lipogenesis in BAs. The decline in DNL enzyme with EC-specific SCF deletion is similar to that observed in c-Kit ablation in BAs, but SCF deletion in ECs does not entirely remove SCF in the plasma due to other origins of SCF, including stromal cells and fibroblasts[62], which results in the maintenance of fasting glucose level and less degree of reduction in DNL enzyme than c-Kit ablation in BAs. Nonetheless, we found that EC-specific SCF deletion prevents the growth of lipid droplets in BAs after thermoneutrality. The growth of lipid droplets in BAs is a process for unilocular lipid droplet formation found in BAT whitening[63]. BAT whitening results in the death of BAs and BAT

inflammation, eventually leading to BAT involution[64]. In this study, we found that SCF/c-Kit signaling is involved in the early phase of lipogenesis and contributes to the BAT whitening process. Thus, inhibition of SCF or c-Kit might be considered as a possible way to prevent lipid accumulation and the resulting BAT whitening.

## Methods

### Study approval

Animal care and experimental procedures were performed under the approval (KA2018-70) of the Institutional Animal Care and Use Committee of Korea Advanced Institute of Science and Technology (KAIST).

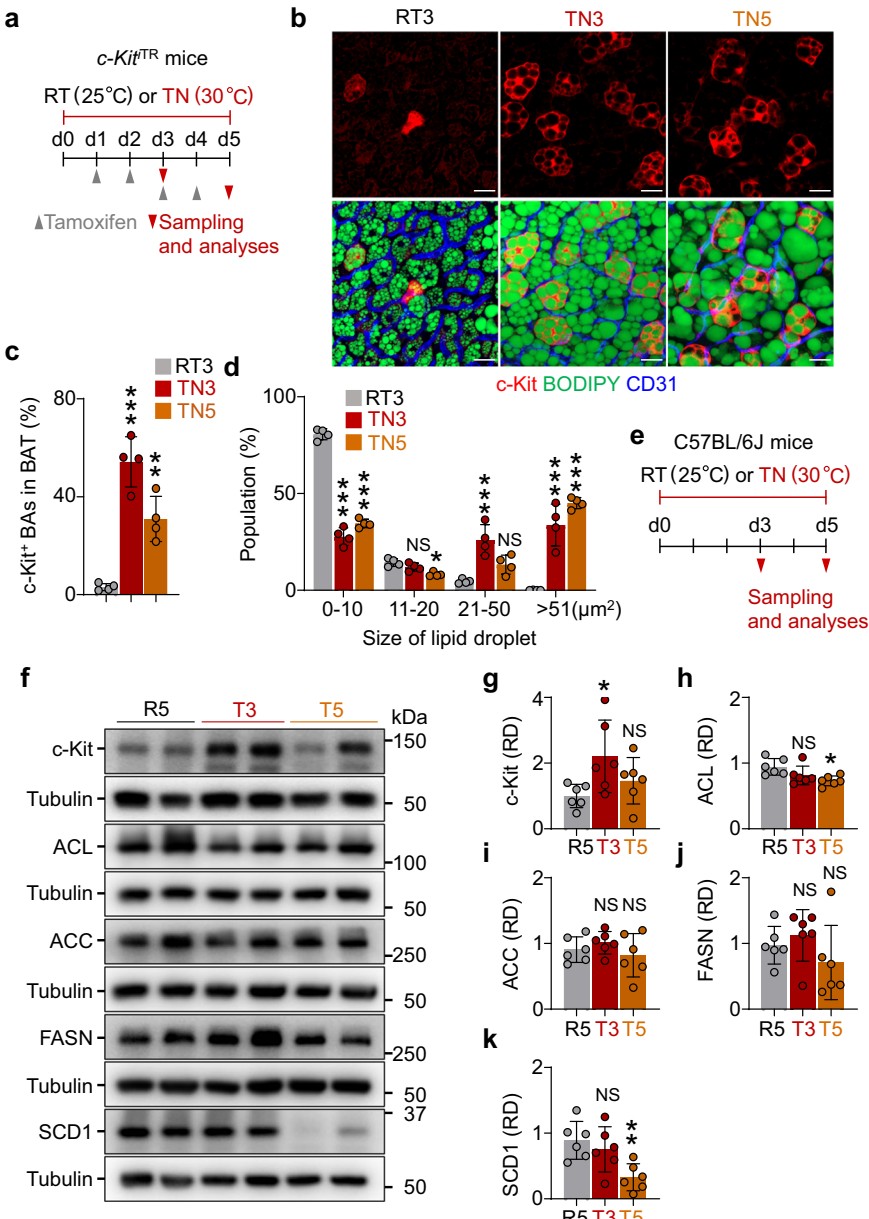

**Fig. 5 | Thermoneutrality increases the number of c-Kit⁺ BAs and expression of c-Kit in BAT. a** Diagram depicting the experimental procedure for the exposure of 8-week-old *c-Kit*^iTR mice to RT (25 °C) or thermoneutrality (TN, 30 °C) for 3 or 5 days with tamoxifen administrations. **b–d** Representative of the number of c-Kit⁺ BAs, and the populations of different sizes of lipid droplets in BATs from the *c-Kit*^iTR mice at RT (25 °C) or thermoneutrality (TN, 30 °C) for 3 or 5 days (RT3, TN3, or TN5). Scale bars, 20 μm. Each dot indicates a value from one mouse and *n* = 4 mice/group from two independent experiments. Vertical bars indicate mean ± SD. ***$P < 0.001$ versus RT by one-way ANOVA test followed by Tukey's post-hoc test (**c**). Each dot indicates the population (%) of the total of 1200–2400 droplets (100%) from three portions

of BAT in one mouse and *n* = 4 mice/group from two independent experiments. Vertical bars indicate mean ± SD. *$P < 0.05$ and ***$P < 0.001$ versus R3 by one-way ANOVA test followed by Tukey's post-hoc test (**d**). NS, not significant. **e** Diagram depicting the experimental procedure for the exposure of 8-week-old C57BL/6 J mice to RT (25 °C) or thermoneutrality (TN, 30 °C) for 3 or 5 days (R5, T3, or T5). **f–k** Representative immunoblotting and comparisons of c-Kit and UCP1 in BATs from C57BL/6 J mice at R5, T3, and T5. Each dot indicates a value from one mouse and *n* = 6 mice/group from two independent experiments. Vertical bars indicate mean ± SD. *$P < 0.05$ versus R5 by one-way ANOVA test followed by Tukey's post-hoc test. NS, not significant. Protein sizes are indicated as kilodalton (kDa).

## Mice and treatment

C57BL/6 J, *tdTomato*^flox/flox (#007909), *c-Kit*^flox/flox (#042035)[53], *UCP1*-Cre (#024670), and *LysM*-Cre (#004781) mice were obtained from The Jackson Laboratory. *Scf*^+/gfp and *Scf*^flox/flox mice were kindly provided by Dr. Sean Morrison (Southwestern University, USA)[36,65]. *c-Kit*-CreER^T2 and *VE-cadherin*-CreER^T2 mice were kindly provided by Dr. Bin Zhou (Chinese Academy of Sciences in Shanghai, China) and Dr. Yoshida Kubota (Keio University, Japan), respectively[66,67]. All mice were bred and maintained under specific pathogen-free conditions at KAIST. Mice were housed under a 12 h light/dark cycle within a temperature-

controlled room (21–22 °C) and allowed to free access food (Teklad global 18% protein rodent diet, #2018C, Envigo®) and water. To induce Cre activity in the CreER^T2 mice, tamoxifen (50 mg/kg of body weight, #T5648, Sigma-Aldrich) was administered by oral gavage at indicated time point. Cre or CreER^T2-negative but flox/flox-positive mice among the littermates were regarded as Control mice. Tamoxifen was given to both the CreER^T2-positive and CreER^T2-negative (Control) mice in the same manner. To avoid the off-target effect of tamoxifen on iWAT browning in the CreER^T2 female mice[68] and to preclude the possible sexual differences among the mice groups, only male mice were used

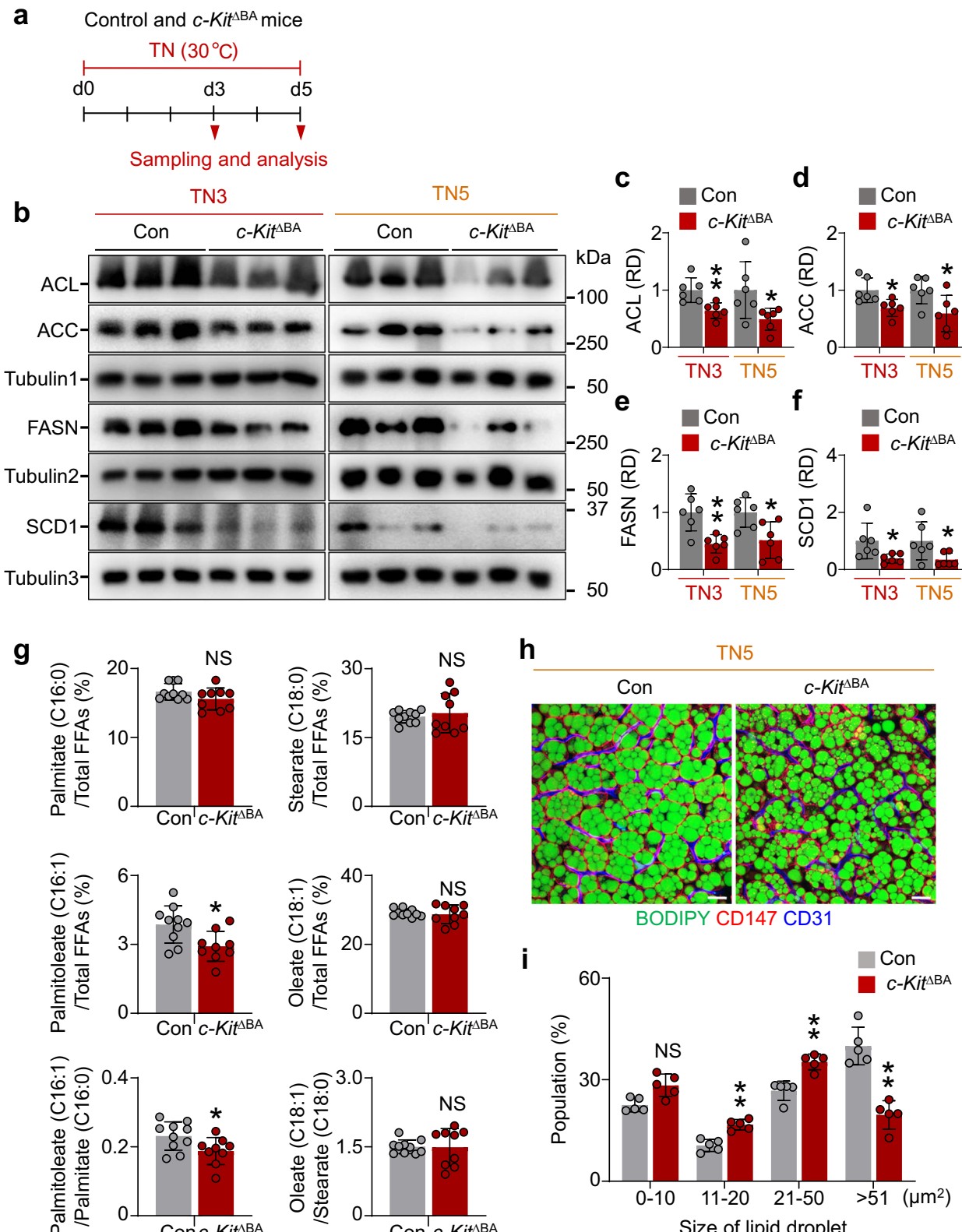

in all of this study. For the β3-AR activation, CL316,247 (1 mg/kg of body weight, #C5976, Sigma-Aldrich) was injected intraperitoneally (i.p.) at the indicated time point. Mice were anesthetized with i.p. injection of a combination of anesthetics (80 mg/kg ketamine and 12 mg/kg xylazine) before any procedure. Mice were euthanized by $CO_2$ inhalation.

**Isolation of nuclei from BAs for single-nucleus RNA sequencing (snRNA seq)**

Interscapular BATs were harvested from 8-week-old C57BL/6 J mice and incubated in RPMI buffer containing 0.1% collagenase type II (#LS004177, Worthington), 0.1% dispase (#17105041, Gibco), and 0.05% trypsin (#T4799, Sigma-Aldrich) for 60 min at 37 °C with

**Fig. 6 | BA-specific c-Kit deletion reduces SCD1 and attenuates the growth of lipid droplets in BAs at thermoneutrality. a** Diagram depicting experimental procedure for the exposure of 8-week-old control (Con) and *c-Kit*<sup>ΔBA</sup> mice to thermoneutrality (TN, 30 °C) for 3 or 5 days, and sampling and analyses.
**b–f** Representative immunoblotting and comparisons of ACL, ACC, FASN, and SCD1 in BAT from Con and *c-Kit*<sup>ΔBA</sup> mice at the thermoneutrality for 3 or 5 days (TN3 or TN5). The same amount of protein loading in each lane is verified by immunoblotting of tubulin1 for ACL and ACC, tubulin2 for FASN, and tubulin3 for SCD1. Each dot indicates a value from one mouse and *n* = 6 mice/group from two independent experiments. Vertical bars indicate mean ± SD. *$*P < 0.05$, $**P < 0.01$ versus Con by two-tailed *t*-test. Protein sizes are indicated as kilodalton (kDa).
**g** Comparisons of the proportions of palmitate (16:0), palmitoleate (16:1), stearate

(C18:0), and oleate (C18:1) in total FFAs of BATs between Con versus *c-Kit*<sup>ΔBA</sup> mice at the thermoneutrality for 3 days. SCD1 desaturation indices calculated by the ratio of 16:1/16:0 and 18:1/18:0 are shown. Each dot indicates a value from one mouse and *n* = 9–10 mice/group. Vertical bars indicate mean ± SD. *$*P < 0.05$ versus Con by two-tailed *t*-test. NS, not significant. **h, i** Representative images and comparisons of the populations of different sizes of lipid droplets in BATs from Con and *c-Kit*<sup>ΔBA</sup> mice at the thermoneutrality for 5 days (TN5). Scale bars, 20 μm. Each dot indicates the population (%) of total 1200–2400 droplets (100%) from three portions of BAT in one mouse and *n* = 5 mice/group from two independent experiments. Vertical bars indicate mean ± SD. $**P < 0.01$ versus Con by two-way ANOVA test followed by Sidak test. NS not significant.

constant shaking at 1000 rpm (Eppendorf thermomixer, Sigma-Aldrich). After inactivating collagenase with RPMI buffer containing 10% fetal bovine serum (FBS) (#26140079, Gibco), the cell suspension was filtered through a 100 μm nylon mesh (BD Biosciences), followed by centrifugation at $100 g$ for 5 min. Because massive amounts of fragmented vasculatures (FVs) were already aggregated with floating BAs after centrifugation, the entire floating cells were digested in Nuclei EZ lysis buffer (#NUC-101, Sigma-Aldrich) to disrupt cell membranes for the isolation of nuclei. The digestion was filtered through a 40 μm nylon mesh, followed by centrifugation at 350 g for 10 min. The pellet was resuspended and incubated with Vybrant™ DyeCycle™ (#V35004, Thermo Fisher Scientific) for 30 min at 4 °C to identify the nuclei and sorted with a BD FACS Aria cell sorter (BD Biosciences) equipped with a 75-mm nozzle. In the total nuclei population detected by SSC-A and FSC-A distribution, doublet nuclei were removed by the gating in FSC-W and FSC-A detectors, and then the Vybrant™ Dye-Cycle™-positive nuclei were sorted (Supplementary Fig. 12). Vybrant™ DyeCycle™-negative cluster was defined as unstained nuclei. Compensation was performed at the time of acquisition in Diva software using compensation beads (#552843, BD Biosciences).

### Single-nucleus library preparation and sequencing

The FACS-assisted isolated nuclei in the mixture of BAs and fragmented vasculature were counted using Countess® II FL Automated Cell Counter (ThermoFisher Scientific), and a total of 20,000 nuclei were loaded on a microwell cartridge of the BD Rhapsody Express system (BD Biosciences). Single-nucleus whole transcriptome libraries were prepared according to the manufacturer's instructions using BD Rhapsody WTA Reagent kit (#633802, BD Biosciences). The final libraries were quantified using a Qubit Fluorometer with the Qubit dsDNA HS Kit (ThermoFisher Scientific), and the size distribution was measured using the Agilent high-sensitivity DNA chip assay on a Tapestation system (Agilent technologies). Lastly, the generated single-nucleus libraries were sequenced by Illumina HiSeq-X platform (150 cycles, Illumina). Raw sequencing data were processed via the standard Rhapsody analysis pipeline (BD Biosciences) on Seven Bridges (https://www.sevenbridges.com) according to the manufacturer's recommendation. The final output of Seven Bridges (molecule per cell matrix) was then analyzed in 'R' (version 4.0.3) using the package Seurat (version 4.0.6). For the pre-processing, we first removed debris and empty droplets by applying DIEM[69] with default options, except for setting the lower bound for the number of detected genes for defining cell-containing droplets as 200[70]. Then, we removed ambient RNA using decontX function in R package "Celda"[71] with default options. From the pre-processed matrix, potential dead cells with more than 50% of UMIs mapped to mitochondrial genes were discarded. After initial clustering, a small number of contaminating cell clusters, such as immune cells, were also removed. In addition, cells simultaneously expressing mutually exclusive marker genes for EC, adipocyte, and stromal cells were considered as doublets and removed. For downstream analysis, the resulting expression matrix after quality control was used. First, normalization was performed by

dividing expression values in a cell by the total expression of the given cell. Then, $log_2$ transformation with the addition of pseudo-count of 1 was performed. After the identification of highly variable genes and scaling, principal component analysis (PCA) and uniform manifold approximation and projection (UMAP) were performed for visualization in 2-dimension plots. Unsupervised clustering was achieved by applying Louvain algorithm. For the integration of datasets, methods implemented in Seurat were used. Briefly, highly variable genes were selected and used for finding the anchors between datasets for integration. Differentially expressed genes were identified using "MAST" test in the R package Seurat with options: min.pct = 0.2, logfc.threshold = 0.1. To infer ligand-receptor interactions, R package CellChat (https://doi.org/10.1038/s41467-021-21246-9) was used. Volcano plots were generated using the EnhancedVolcano 'R' package (https://doi.org/10.18129/B9.bioc.EnhancedVolcano).

### Sample preparations for histological analyses and immunofluorescence staining

After anesthesia, mice were perfused with ice-cold PBS followed by 2% formalin (#HT501128, Sigma-Aldrich) through the left ventricle by puncturing the right auricle. BAT, WATs, liver, skeletal muscle, and pancreas were harvested, and the tissues were then post-fixed in 2% formalin at 4 °C overnight. The liver, skeletal muscle, and pancreas were cut into 200 μm-thick sections using a vibrating microtome (#VT1200S, Leica). ATs were whole-mounted. The samples were washed with PBS several times and blocked with 5% goat or donkey serum (Jackson ImmunoResearch) in 0.2% Triton-X 100 in PBS (PBST) for 1 h at RT. The samples were then incubated with the following primary antibody diluted in the blocking solution at 4 °C overnight: anti-CD31 (hamster monoclonal, #MAB1398Z, Millipore); anti-GFP (goat polyclonal, #ab6658, Abcam); anti-F4/80 (rat monoclonal, #MCA497G, Bio-Rad); Alexa Fluor™ 647 conjugated anti-phalloidin [mouse, #8940, Cell Signaling Technology (CST)]; PE-conjugated CD147 (rat monoclonal, #123707, BioLegend). After several washes with PBS, the samples were incubated at 4 °C for 4 h with Cy3-, Cy5-, or Alexa488-conjugated secondary antibodies (Jackson ImmunoResearch) in PBS. Neutral lipids in adipocytes were stained with BODIPY™ 493/503 (#D3922, Invitrogen) or LipidTOX red dyes (#H34476, Invitrogen) according to the manufacturer's instructions. To visualize lipid droplets, the cells were fixed with 1% formalin for 1 h at RT, then washed with PBS several times and blocked with 5% goat or donkey serum in 0.2% Triton-X 100 in PBS (PBST) for 1 h at RT. Cells were then incubated with LipidTox (0.2 μg/ml) diluted in the blocking solution at 4 °C overnight. Lipid accumulation in the cell was calculated by determining the total LipidTox⁺ lipid area per total GFP⁺ cell area, and the data were presented as a percentage.

### Imaging and morphometric analyses

Immunofluorescent images were acquired using an LSM880 confocal microscope (Carl Zeiss). ZEN 2.3 software (Carl Zeiss) was used to acquire and process images. Confocal images of stained samples were processed with maximum intensity projections of single plane z-stack

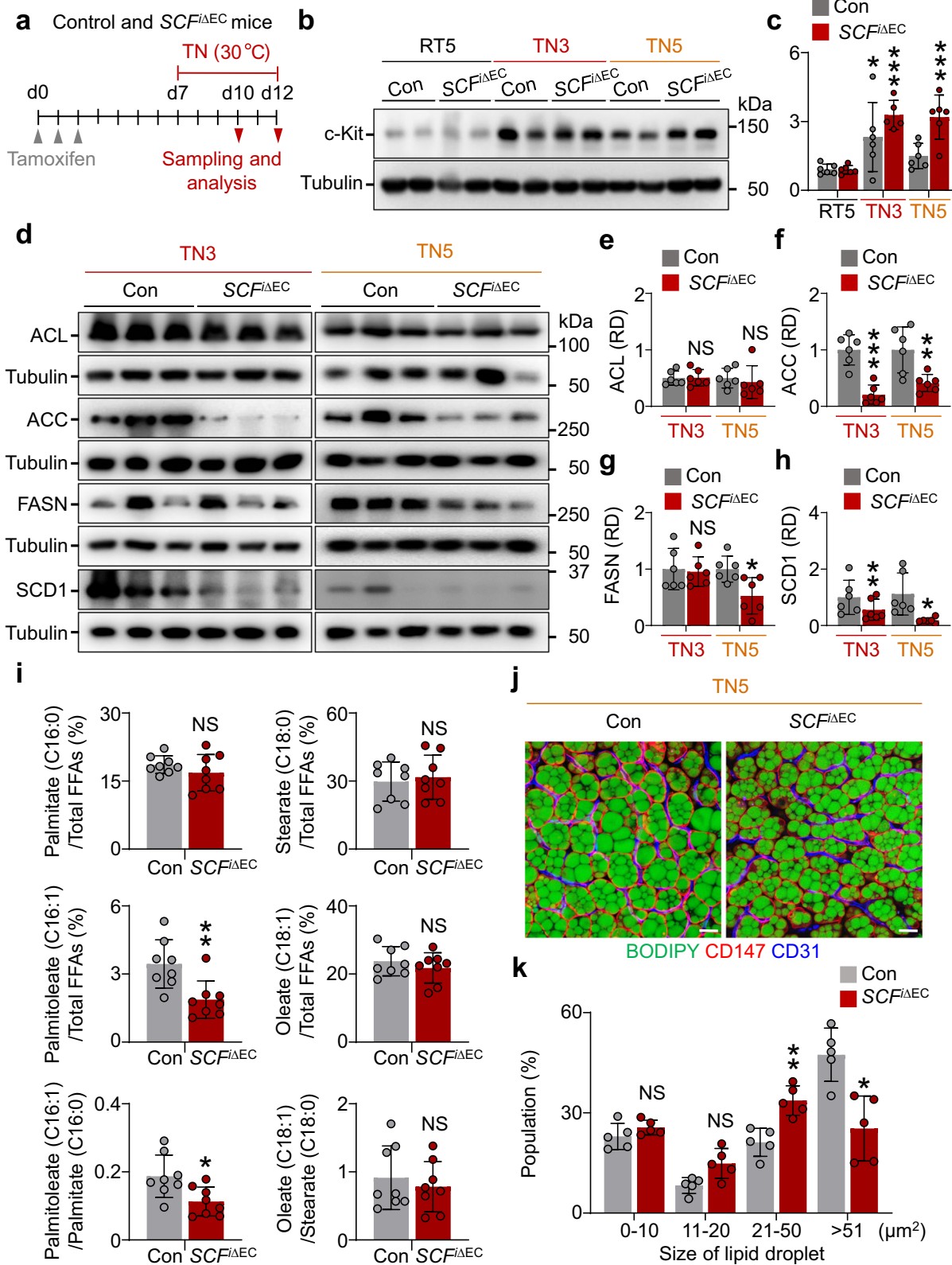

images through the 4–10 μm thickness of tissues, which were all taken at a resolution of 1024 X1024 pixels with the LD C-Apochromat 40x/1.1 NA water immersion Corr M27 (LSM 880) with multichannel scanning in the frame. 3D reconstruction images were created from z-stack confocal images using the 3D tab in ZEN 2.3 software. Images were obtained from 3 different portions of each sample, and morphometric measurements were performed in 210 μm × 210 μm fields of an image

using ImageJ software (version 1.8.0_172, NIH) and Zen 2.3 software. The vessel coverage of SCF was defined as the total SCF-GFP[+] area in CD31[+] ECs divided by the total area of CD31[+] ECs. The macrophage coverage of SCF was defined as the total SCF-GFP[+] area in F4/80[+] macrophages divided by the total area of F4/80[+] macrophages. The total number of c-Kit[+] BAs in tdTomato reporter mouse was counted manually and then presented as a percentage of total BAs in 3

**Fig. 7 | EC-specific SCF deletion reduces SCD1 and attenuates the growth of lipid droplets in BAs at thermoneutrality. a** Diagram depicting an experimental procedure for the exposure of 8-week-old control (Con) and $SCF^{i\Delta EC}$ mice to thermoneutrality (TN, 30 °C) for 3 or 5 days, and sampling and analyses. **b,c,** Representative immunoblotting and comparison of c-Kit in BAT in Con and $SCF^{i\Delta EC}$ mice at RT or thermoneutrality for 3 or 5 days (RT5, TN3, or TN5). The same amount of protein loading in each lane is verified by immunoblotting of tubulin. Each dot indicates a value from one mouse and $n = 5$ mice for $SCF^{i\Delta EC}$ group in TN3 and $n = 6$ mice for other groups from two independent experiments. Vertical bars indicate mean ± SD. *$P < 0.05$, ***$P < 0.001$ versus RT by one-way ANOVA test followed by Tukey's post-hoc test. Protein sizes are indicated as kilodalton (kDa). **d–h** Representative immunoblotting and comparisons of ACL, ACC, FASN, and SCD1 in BAT of Con and $SCF^{i\Delta EC}$ mice at thermoneutrality for 3 or 5 days (TN3 or TN5). The same amount of protein loading in each lane is verified by immunoblotting of tubulin. Each dot indicates a value from one mouse and $n = 6$ mice/group from two independent experiments.

Vertical bars indicate mean ± SD. *$P < 0.05$, **$P < 0.01$, ***$P < 0.001$ versus Con by two-tailed t-test. NS, not significant. Protein sizes are indicated as kilodalton (kDa). **i** Comparisons of the proportions of palmitate (16:0), palmitoleate (16:1), stearate (C18:0), and oleate (C18:1) in total FFAs of BATs between Con versus $SCF^{i\Delta EC}$ mice at the thermoneutrality for 3 days. SCD1 desaturation indices calculated by the ratio of 16:1/16:0 and 18:1/18:0 are shown. Each dot indicates a value from one mouse and $n = 8$ mice/group. Vertical bars indicate mean ± SD. *$P < 0.05$ versus Con by two-tailed t-test. NS, not significant. **j, k** Representative images and comparison of the populations of different sizes of lipid droplets in BAT from Con and $SCF^{i\Delta EC}$ mice at the thermoneutrality for 5 days (TN5). Scale bars, 20 µm. Each dot indicates the population (%) of a total of 1200–2400 droplets (100%) from three portions of BAT in one mouse and $n = 5$ mice/group from two independent experiments. Vertical bars indicate mean ± SD. *$P < 0.05$ versus Con by two-way ANOVA test followed by Sidak test. NS not significant.

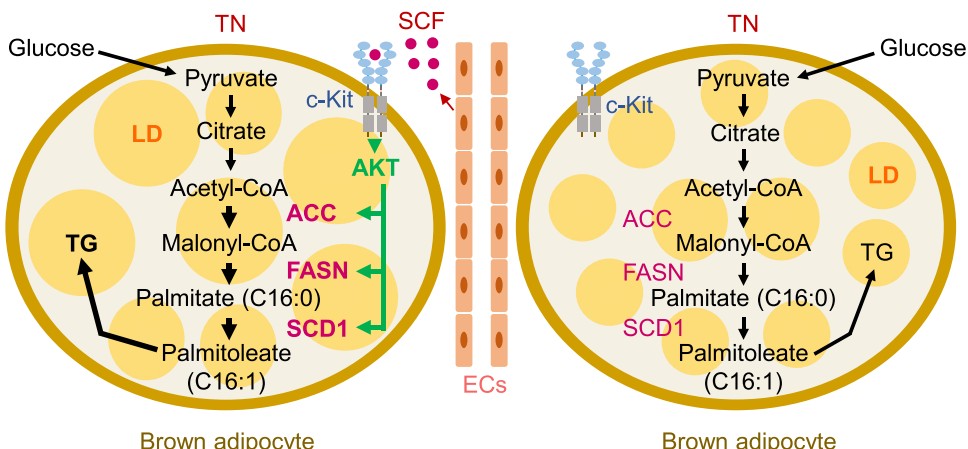

**Fig. 8 | Schematic diagram depicting the promoting role of SCF/c-Kit signaling in lipid accumulation in BAs.** Activation of c-Kit in BAs by SCF derived from ECs increases protein levels of DNL enzymes, which enhances the growth of lipid droplets. TN thermoneutrality, TG triglycerides, LD lipid droplet.

independent images. The size of a single brown adipocyte was analyzed by measuring the area of CD147+ BAs manually using the polygon tool and then presented as the average size of 90–120 cells in a BAT from one mouse. The size (area; µm²) of a single lipid droplet was defined as an area of BODIPY+ lipid droplet. The size of 1200–2400 droplets obtained from one BAT was categorized into size-based 4 subpopulations [small (0–10 µm²), middle (12–20 µm²), large (21–50 µm²), and larger (> 51 µm²)]. The data was presented as a percentage of the total number of lipid droplets in three independent images. Lipid droplet in the cultured BAs was visualized by the LipidTox staining, calculated the lipid accumulation in the cell by total LipidTox+ lipid area per total GFP+ cell area, and presented the data as a percentage.

## Immunoblotting assay

For the protein isolation, frozen tissues in RIPA buffer (#CBR002, LPS solution) containing protease inhibitor and phosphatase inhibitor (#5872, CST) were homogenized with ULTRA-TURRAX® dispersers (IKA) at 20,000 rpm for 2–3 min at 4 °C until tissues were completely lysed, followed by centrifugation at 16,000 g for 10 min at 4 °C. Total protein isolated from each sample was measured and normalized with a Pierce BCA Protein Assay Kit (#23225, ThermoFisher Scientific). An equal amount of protein from tissue lysate was loaded into each well of a 7.5–12% SDS-polyacrylamide gel after denaturation with SDS loading buffer for 5 min at 95 °C. After electrophoresis, proteins were transferred to a PVDF membrane, incubated with 2% bovine serum albumin (BSA) blocking buffer for 1 h at RT, and blotted with the following antibodies overnight: rabbit anti-c-Kit (#3074); rabbit anti-ACC (#3676); rabbit anti-FASN (#3180); rabbit anti-SCD1 (#2794); rabbit

anti-GAPDH (#5174) (CST); rabbit anti-p-AKT1 (ser473) (#9018); rabbit-anti-AKT1 (#2938); rabbit-anti-p-AKT2 (Ser474) (#8599); rabbi-anti-AKT2 (#3063); rabbi-anti-p-p44/42 MAPK (Thr202/Tyr204) (#4377); rabbi-anti-p44/42 MAPK (#4695); rabbit anti-PGC1-α (#ab191838); rabbit anti-tyrosine hydroxylase (#ab137869); rabbit anti-ACL (#ab40793); rabbit anti-UCP1 (#ab23841) (Abcam) (4 h at RT or overnight); rat anti-α-tubulin (#sc-69970, Santacruz). After several washes with TBST solution (#CBT007, LPS solution), the membrane was incubated with anti-rabbit (#7074, CST) or anti-mouse (#7076, CST) secondary peroxidase-conjugated antibody for 1 h at RT. Target proteins were detected using ECL western blot detection solution (#WBKLS0500, Millipore). The same amount of protein loading in each lane is verified by immunoblotting of tubulin or GAPDH. Images were obtained using Amersham Imager 600 (GE Healthcare Life Sciences), and protein density was measured using Image Studio Lite (LI-COR).

## Unilateral denervation of sympathetic nerves on BATs

Unilateral denervation on BAT was performed in the 8-week-old mice as previously described[10,45]. After anesthesia, the mouse was weighed, shaved, and placed on a warm surgical table. Under a stereomicroscope, interscapular BAT lobes were elevated from the underlying cervical and scapular caudally to separate them from the surrounding muscle and expose sympathetic nerves. Five branches of intercostal nerves of the left lobe were isolated and cut (denervated) without disrupting the vasculature, and the right intercostal nerve bundles were left intact as an internal negative control. After the unilateral denervation UDN), the interscapular BAT lobes were returned to their original position, and the skin was sutured with a tissue adhesive and skin clips (#RS-9255, Roboz Surgical Instrument). The operated mice

were housed individually in a clean cage. BATs were isolated at the indicated time points and frozen at −80 °C until further analysis.

### BAs isolation and immortalization

Interscapular BATs were isolated at postnatal day 2 in C57BL/6 J mice, minced, and digested with isolation buffer (2 mg of collagenase in 2 ml of isolation buffer containing 0.123 M NaCl, 5 mM KCl, 1.3 mM $CaCl_2$, 5 mM glucose, 100 mM HEPES (4-(2-hydroxyethyl) piperazine-1-ethanesulfonic acid), and 4% BSA) for 40 min at 37 °C with constant shaking at 900 rpm (Eppendorf thermomixer, Sigma-Aldrich). The digestion was quenched with DMEM/F12 containing 10% FBS, and the dissociated cells were passed through a 100 µM filter, followed by centrifugation at 100 $g$ for 10 min. The pellet containing precursor cells was washed once with isolation buffer and centrifuged again. The resulting pellet was resuspended in high glucose Dulbecco's modified Eagle medium (DMEM) media containing 10% FBS, seeded on a 35 mm plate, and grown in a humidified atmosphere of 5% $CO_2$ and 95% air. The medium was changed every other day. After reaching 80% confluence, cells were passed to 10 cm plates and infected with the puromycin resistance retroviral vector pBabe encoding SV40 T antigen (Addgene) for 24 h. Following infection, the brown adipose precursor cells were maintained in a culture medium for 72 h and then subjected to selection with puromycin (1 mg/ml) for two weeks.

### c-Kit overexpression in immortalized pre-adipocytes and differentiation

For the overexpression of *c-Kit*, mouse *c-Kit* cDNA (#MR227469, Ori-Gene) was inserted in FUGW lentiviral plasmid (#14883, Addgene) tagging with GFP, and the resulting plasmid was transfected into pre-adipocytes for 24 h. As a control, the empty FUGW lentiviral plasmid was transfected into pre-adipocytes in the same manner. GFP-positive Control or *c-Kit* overexpressed pre-adipocytes were sorted with a BD FACS Aria cell sorter. The sorted preadipocytes were grown in DMEM media until reached around 95% confluency. The cells were incubated in differentiation media (high glucose DMEM media supplemented with 10% FBS, 20 nM insulin (#I5523, Sigma-Aldrich) and 1 nM T3 (#T2877, Sigma-Aldrich) for 48 h and then incubated in induction media that was further supplemented with 0.5 mM isobutylmethylxanthine (#I5879, Sigma-Aldrich), 0.5 mM dexamethasone (#D1756, Sigma-Aldrich), and 0.125 mM indomethacin (#I7378, Sigma-Aldrich) in differentiation media for four days until exhibiting a fully differentiated phenotype with accumulation of multilocular lipid droplets. BSA (100 ng/ml, #23209, Thermo Fisher Scientific), VEGF-A (100 ng/ml, #493-MV, R&D Systems), or SCF (100 ng/ml, #455-MC, R&D Systems) was treated as indicated. For inhibitor treatment, cells were pretreated for 60 min with dynasore (20 µM, #324410, Sigma-Aldrich), wortmannin (30 nM, #W1628, Sigma-Aldrich), AKT1/2 inhibitor (25 nM, #A6730, Sigma-Aldrich), PD184352 (1 µM, #PZ0181, Sigma-Aldrich), SU-5402 (1 µM, #SMl0443, Sigma-Aldrich), Y-27632 (10 µM, #Y0503, Sigma-Aldrich), or ruxolitinib (10 µM, #7064, TOCRIS), followed by SCF treatment for 12 h. All experiments were performed within five passages following *c-Kit* overexpression.

### Real-time PCR

Total RNA was extracted using TRIzol isolation reagent (Invitrogen, Carlsbad, CA, USA) under the manufacturer's instructions. RNA concentration was spectrophotometrically determined using NanoDrop (Thermo Fisher Scientific, Waltham, MA, USA). Using murine leukemia virus reverse transcriptase and oligo (dT)16 primer, two micrograms of cell RNAs were reverse transcribed. The resulting cDNAs from samples were assayed in duplicate. Reverse transcription-polymerase chain reaction was conducted using 2X SYBR green PCR master mix on a real-time PCR system (Applied Biosystems, Waltham, MA, USA). Gene expression data were normalized to the housekeeping gene *GAPDH* and analyzed using the delta-delta cycle threshold method (ΔΔCt).

Primer sets were designed using PrimerQuest tool (Integrated DNA Technologies); *Ucp1* (5′-GAGGTCGTGAAGGTCAGAATG-3′, 5′-AAGCTT TCTGTGGTGGCTATAA-3′) *Gapdh* (5′-AATGTGTCCGTCGTGGATCT-3′, 5′-CCTGTTGCTGTAGCCGTATT-3′).

### Transmission electron microscopy (TEM)

The BATs were fixed with 2% glutaraldehyde-2% paraformaldehyde in a 0.1 M sodium cacodylate buffer pH 7.4 for 24 h at 4 °C. The fixed samples were washed with 0.1 M sodium cacodylate and then post-fixed in 1% osmium tetroxide (EMS) for 1 h at RT. After several washes in 0.1 M sodium cacodylate, the samples were dehydrated in a series of graded ethanol series, substituted with propylene oxide, and then progressively infiltrated by a 2:1, 1:1, and 1:2 mixture of propylene oxide and embedded in EMBed 812 resin (EMS). Polymerization was performed at 60 °C for 40 h. Ultrathin (70 nm) sections were prepared using an ultramicrotome (Leica, EM UC7) collected on 100-mesh formvar/carbon-coated copper grids (EMS). The grids were stained with 2 % uranyl acetate (7 min) and Reynold's lead citrate (3 min). The sections were examined with a transmission electron microscope (FEI, Tecnai G2 spirit TWIN) at 120 kV accelerating voltage. Images were acquired using a NanoSprint12 CMOS camera (Advanced Microscopy Techniques). TEM was carried out in EM & Histology Core Facility at the BioMedical Research Center in KAIST.

### Indirect respiration calorimetry measurement

The calorimetry measurements were performed as previously described[72]. The Oxymax/CLAMS calorimetry system (Columbus Instruments) was maintained on a 12 h light/dark cycle at 25 °C in the core facility at the BioMedical Research Center (KAIST). Mice were acclimated to the core facility for 24 h and then transferred to calorimetry chambers contained in a 25 °C incubator for another 24 h acclimation. The calorimetry measurement was maintained for 48 h. Room air was drawn through the calorimetry chambers at 500 ml/min. Samples of dried room and chamber air were analyzed for oxygen and carbon dioxide content using the Oxymax system. Calorimeter calibration was performed daily before the beginning of the measurement. A 0.50% $CO_2$ and 20.50% $O_2$ (balance nitrogen) calibration gas and dry room air were used for the calibration of the analyzers.

### Measurement of SCF in ATs and serum

Total proteins from the adipose tissues were extracted and normalized as described above (**Immunoblotting assay, Methods**). Plasma was isolated from the blood collected by penetrating the retro-orbital sinus in mice with a sterile hematocrit capillary coated with EDTA. The concentration of SCF in the adipose tissues or plasma was measured using a Mouse SCF Quantikine ELISA Kit (#MCK00, R&D systems) according to the manufacturer's instructions.

### Fasting glucose measurement

Tamoxifen was treated in 7-week-old control and *Scf*^iΔEC mice by oral gavage for 3 consecutive days. For the assessment of fasting glucose, *Scf*^iΔEC mice fasted for 12 h, and then a drop of blood was collected from a tiny cut on the skin over the tail vein with a surgical blade. Tail blood glucose concentrations were measured with a glucometer (Accu-check, Roche) once a week from 8 to 11 weeks of age.

### UPLC − QTOF − MS analysis

To extract lipids, 40 mg of BAT was mixed with 600 µl of $H_2O$-methanol (1:1, v/v) and homogenized twice with zirconium oxide beads at 5000 rpm for 20 seconds using a Precellys 24 tissue grinder (Bertin Technologies, Ampère Montigny-le-Bretonneux, France). After mixing with 600 µl of chloroform and storing at 4 °C for 10 min, the sample was then centrifuged for 20 min at 15,000 $g$ at 4 °C, and 100 µl of the lower phase (chloroform-methanol extract) was dried under a nitrogen stream. The lipid extract was reconstituted with 200 µl of

internal standard SPLASH LipidoMix (Part No. 330707, Avanti Polar Lipids inc., Alabaster, AL, USA) in isopropanol and diluted with $H_2O$-isopropanol (2:8, v/v). Finally, 5 µl of each sample was injected into ultra-performance liquid chromatography (UPLC) coupled with quadrupole time-of-flight mass spectrometry (QTOF MS). Lipid extracts were separated using an Acquity UPLC system (Waters Corp., Milford, MA, USA) equipped with an Acquity UPLC CSH C18 column (100 mm × 2.1 mm, 1.7 µm particle size; Waters Corp.) at 55 °C and a flow rate of 0.4 ml/min. The mobile phase consisted of 10 mM ammonium formate in $H_2O$-acetonitriler mixture (4:6, v/v) containing 0.1% formic acid (solvent A), and isopropanol-acetonitrile mixture (9:1, v/v) containing 0.1% formic acid (solvent B) for positive ion mode. The mobile phase for negative ion mode comprised 10 mM ammonium acetate in $H_2O$-acetonitriler mixture (6:4, v/v) (solvent A) and isopropanol-acetonitrile mixture (9:1, v/v) (solvent B). The steps of the gradient were programmed as follows: 40–43% B from 0 min to 2 min, 43–50% B from 2 min to 2.1 min, 50–54% B from 2.1 min to 12 min, 54–70% B from 12 min to 12.1 min, 70–99% B from 12.1 min to 18 min, 99–40% B from 18 min to 18.1 min, and 40% B for 2 min to equilibrate for the next run. A Waters Xevo G2-XS QTOF MS (Waters Corp., Milford, MA, USA) was used to acquire mass spectral data in positive and negative ESI modes with an *m/z* range of 80–1500. The following parameter settings were used: capillary voltages of +2000 and −1000 V for positive and negative ion modes, a sample cone of 30 V, source offset of 80 V, source temperature of 120 °C, desolvation temperature of 600 °C, cone gas flow of 150 and 50 L/h for positive and negative ion mode. To identify lipids in BAT, the MS/MS data were acquired by product ion scan mode, in which the peaks for target lipids such as acylcarnitines (ACars), free fatty acid (FFAs), lysophosphatidylethanolamines (LysoPEs), lysophosphocholines (LysoPCs), phosphatidylcholines (PCs), phosphatidylethanolamines (PEs), phosphatidylglycerols (PGs), phosphatidylinositols (PIs), diacylglycerols (DAGs) and triacylglycerols (TAGs) were fragmented. Leucine-enkephalin was used as the lock mass to ensure mass accuracy generating an [M + H]$^+$ ion at *m/z* 556.2771 in positive mode, and an [M-H]$^−$ ion at *m/z* 554.2771 in negative mode. All the samples were pooled in equal amounts to generate quality control (QC) samples, which were analyzed prior to running sample acquisition and after every seven samples to monitor the stability and reproducibility of the analytical system. Progenesis QI software (Waters, Milford, MA) was used to perform peak finding, alignment, and generating peak tables of *m/z* and retention times (min). And then, the intensity of all peaks was normalized using the total sum of peak intensities for positive mode and batch normalization using QC samples for negative mode[73]. We identify lipids using our in-house library.

## Statistics and reproducibility
No statistical methods were used to predetermine the sample size. The experiments were randomized, and investigators were blinded to allocation during experiments and outcome analyses. All values are presented as mean ± standard deviation (SD). Statistical significance was determined by the two-tailed unpaired t-test between two groups, the one-way ANOVA test followed by Tukey's *post-hoc* test for multiple-group comparison, or the two-way ANOVA test followed by Sidak test for the comparison of multiple-group with two independent variables. Statistical analysis was performed using Prism 7.0 (GraphPad Software). Statistical significance was set to *P*-value < 0.05.

## Reporting summary
Further information on research design is available in the Nature Portfolio Reporting Summary linked to this article.

## Data availability
Single-cell RNA sequencing data are available in National Center for Biotechnology Information's Gene Expression Omnibus under accession number GSE207096. The remaining data are available within the Article or Additional Information. Source data are provided as a Source Data file. Further information and requests for resources and reagents should be directed to and will be fulfilled by Hyuek Jong Lee (hyuekjong.lee@gmail.com) and Gou Young Koh (gykoh@kaist.ac.kr). Source data are provided with this paper.

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

## Acknowledgements

The authors thank Sujin Seo and Junho Jung (IBS) for their technical assistance and Dr. Yongsuk Hur at EM & Histology Core Facility, the BioMedical Research Center, KAIST for the scientific and technical support. We also thank Dr. Sean Morrison (Southwestern University, USA) for *Scf*$^{+/gfp}$ and *Scf*$^{flox/flox}$ mice, Dr. Bin Zhou (Chinese Academy of Sciences in Shanghai, China) for *c-Kit*-CreER$^{T2}$ mice, and Dr. Yoshida Kubota (Keio University, Japan) for *VE-cadherin*-CreER$^{T2}$ mice. This study was supported by the Institute for Basic Science (IBS-R025-D1-2015 to G.Y.K) funded by the Ministry of Science, ICT and Future Planning, Korea.

## Author contributions

H.J.L. and Y.C.K. designed, organized, and performed all experiments, generated the figures, and wrote the paper. M.J.Y. and J.M.K. performed snRNA seq and analyzed the datasets. J.L. and G.H. performed UPLC – QTOF – MS analysis. S.P.H. provided technical support. H.J.L. and G.Y.K. designed, organized, supervised the project, and wrote the paper.

## Competing interests

The authors declare no competing interests.
