## [Peer Review File · Nature Communications]

Endothelial cell-derived stem cell factor promotes lipid accumulation through c-Kit-mediated increase of lipogenic enzymes in brown adipocytesEditorial Note: This manuscript has been previously reviewed at another journal that is not operating a transparent peer review scheme. This document only contains reviewer comments and rebuttal letters for versions considered at *Nature Communications*.

REVIEWER COMMENTS

Reviewer #1 (Remarks to the Author):

This manuscript describes the role of SCF in lipogenesis. While the authors have done a substantial amount of work, the novelty and importance are modest. Given that SCF appears to only regulate lipogenic gene expression and not DNL, is not sufficiently interesting for this audience and more appropriate for a more specialized journal. As brought up by the prior reviewers, there is a large amount of work done in this area where this paper does not stand out.

The work is not entirely supportive of the conclusions, but there are no flaws in the actual analyses that are presented. To claim DNL, the authors need to perform DNL assays.

I have carefully read the responses to the previous concerns and have the following comments when the authors make changes to make this work publishable:

Abstract: It is unclear why the role of EC and lipid accumulation in adipose tissue needs to be studied. What is the rationale for starting this line of study?

The title is not representative of the presented data because no lipogenesis/DNL is not actually shown. The title should be changed to lipogenic gene expression, not DNL or lipogenesis.

The short title SCF/c-Kit signaling regulates lipogenesis in brown adipocytes needs to be corrected as well

Line 41: promotes DNL is incorrect, change to promote lipogenic gene expression

Line 96 change to promote lipogenic gene expression

Line 97 change to promote lipogenic gene expression

Line 98 change to promote lipogenic gene expression

The role in lipid droplet biology in Fig 7 seems robust, but no mechanism is provided. SCD expression is proposed, but that cannot explain the lipid droplet phenotype.

Line 117: this sentence is unclear, please revise.

Given that the reciprocal selectivity of SCF/c-Kit expressions was higher than that of Efnb1/Efnb2/Ephb2/Ephb4 or Bmp1/Bmp6/Bmpr1a/Bmpr2 expressions in ECs versus BAs (Fig. 1f), we chose and focused on investigating the roles of SCF/c-Kit signaling in the BAT.

Reviewer #2 (Remarks to the Author):

The authors have addressed all the previous concerns.

Reviewer #3 (Remarks to the Author):

The manuscript has greatly improved, and author have answered most of my questions. However, I still have some minor comments which should be addressed.

Regarding the second snRNAseq experiment (Extended Data Fig. 6). What did the authors sequence this time? Only BA or was it again the floating fraction? If so, why there are no ECs or PCs in this new dataset? This should be clarified.

There are also some minor spelling mistakes:

- Line 240: BSA (100ng/ml) instead of BAS (100ng/ml)
- Line 468: Tamoxifen instead of Tomoxifen
- Line 498: "MJ recheck this" must be removed

Responses to the reviewers' comments

We appreciate the reviewers for their thoughtful, critical, and constructive comments, which have provided valuable opportunities to improve our work. We have revised the manuscript to address the issues raised by the reviewers.

Reviewer #1 (Remarks to the Author):

This manuscript describes the role of SCF in lipogenesis. While the authors have done a substantial amount of work, the novelty and importance are modest. Given that SCF appears to only regulate lipogenic gene expression and not DNL, is not sufficiently interesting for this audience and more appropriate for a more specialized journal. As brought up by the prior reviewers, there is a large amount of work done in this area where this paper does not stand out. The work is not entirely supportive of the conclusions, but there are no flaws in the actual analyses that are presented. To claim DNL, the authors need to perform DNL assays. I have carefully read the responses to the previous concerns and have the following comments when the authors make changes to make this work publishable:

We agree with this critical comment on our claims without the definitive *in vivo* lipogenesis assay using a radioisotope. Therefore, we made changes in the statements of “*de novo* lipogenesis” and modified the claims and conclusions throughout the manuscript.

Abstract: It is unclear why the role of EC and lipid accumulation in adipose tissue needs to be studied. What is the rationale for starting this line of study?

(revised abstract) Active thermogenesis in the brown adipose tissue (BAT) facilitating the utilization of lipids and glucose is critical for maintaining body temperature and reducing metabolic diseases, whereas inactive BAT accumulates lipids in brown adipocytes (BAs), leading to BAT whitening. Although cellular crosstalk between endothelial cells (ECs) and adipocytes is essential for the transport and utilization of fatty acid in BAs, the angiocrine roles of ECs mediating this crosstalk remain poorly understood. Using single-nucleus RNA sequencing and knock-out mice, we demonstrate that stem cell factor (SCF) derived from ECs upregulates gene expressions and protein levels of the enzymes for *de novo* lipogenesis, and promotes lipid accumulation by activating c-Kit in BAs. In the early phase of lipid accumulation induced by denervation or thermoneutrality, transiently expressed c-Kit on BAs increases the protein levels of the lipogenic enzymes via PI3K and AKT signaling. EC-specific SCF deletion and BA-specific c-Kit deletion attenuate the induction of the lipogenic enzymes and suppress the enlargement of lipid droplets in BAs after denervation or thermoneutrality. These data provide insight into SCF/c-Kit signaling as a novel regulator that promotes lipid accumulation through the increase of lipogenic enzymes in BAT when thermogenesis is inhibited.

The title is not representative of the presented data because no lipogenesis/DNL is not actually shown. The title should be changed to lipogenic gene expression, not DNL or lipogenesis.

(revised title) Endothelial cell-derived stem cell factor promotes lipid accumulation through c-Kit-mediated increase of lipogenic enzymes in brown adipocytes

The short title SCF/c-Kit signaling regulates lipogenesis in brown adipocytes needs to be corrected as well

(revised short title) SCF/c-Kit signaling regulates lipid accumulation in brown adipocytes

Line 41: promotes DNL is incorrect, change to promote lipogenic gene expression

Line 96 change to promote lipogenic gene expression

Line 97 change to promote lipogenic gene expression

Line 98 change to promote lipogenic gene expression

Because we measured not only gene expressions but also protein levels of the enzymes related to DLN, we changed and modified the statements accordingly in these and other sentences throughout the manuscript.

The role in lipid droplet biology in Fig 7 seems robust, but no mechanism is provided. SCD expression is proposed, but that cannot explain the lipid droplet phenotype.

As described in the text, we attributed the substantial attenuation of the induction of lipogenic enzymes ACC, FASN, and SCD1 to the significant attenuation of the growth of lipid droplets in the *SCF^{ΔEC}* mice at thermoneutrality.

Line 117: this sentence is unclear, please revise.

Given that the reciprocal selectivity of SCF/c-Kit expressions was higher than that of *Efnb1/Efnb2/Ephb2/Ephb4* or *Bmp1/Bmp6/Bmpr1a/Bmpr2* expressions in ECs versus BAs (Fig. 1f), we chose and focused on investigating the roles of SCF/c-Kit signaling in the BAT.

We rephrased the sentence to be clear in the revised manuscript.

(p 5) Since the selective expressions of *SCF* and *c-Kit* between ECs versus BAs were distinctively higher than those of *Efnb1/Efnb2* and *Ephb2/Ephb4* or *Bmp1/Bmp6* and *Bmpr1a/Bmpr2* (Fig. 1f), we chose and focused on investigating the roles of SCF/c-Kit signaling in the BAT.

Reviewer #2 (Remarks to the Author):

The authors have addressed all the previous concerns.

Thank you for this encouraging and favorable comment.

Reviewer #3 (Remarks to the Author):

The manuscript has greatly improved, and author have answered most of my questions. However, I still have some minor comments which should be addressed.

Regarding the second snRNAseq experiment (Extended Data Fig. 6). What did the authors sequence this time? Only BA or was it again the floating fraction? If so, why there are no ECs or PCs in this new dataset? This should be clarified.

For the second snRNA-seq experiment (Extended Data Fig. 6), we performed snRNA-seq in the floating mixture of BAs and fragmented vasculature isolated from the D-BAT3. This experiment aimed to elucidate the roles of c-Kit in BAs by comparing differentially expressed genes between *c-Kit*⁺ BAs versus *c-Kit*⁻ BAs. That was why we focused on analyzing and comparing only the BA clusters among the three clusters obtained.

(p 8) To uncover the roles of c-Kit in BAs, we performed snRNA-seq in the floating layer of D-BAT at day 3 after denervation (**Extended Data Fig. 6a**) and focused on analyzing only the BAs cluster among the three clusters obtained (**Extended Data Fig. 6b**).

There are also some minor spelling mistakes:

-Line 240: BSA (100ng/ml) instead of BAS (100ng/ml)

-Line 468: Tamoxifen instead of Tomoxifen

-Line 498: "MJ recheck this" must be removed

Thank you for pointing those out. We corrected typo errors throughout the manuscript.

REVIEWERS' COMMENTS

Reviewer #1 (Remarks to the Author):

The authors have responded to all my concerns.

Reviewer #3 (Remarks to the Author):

I do not have further comments.

Responses to the reviewers' comments

We appreciate the reviewers for their favorable and encouraging comments.

Reviewer #1 (Remarks to the Author): The authors have responded to all my concerns.

Thank you.

Reviewer #3 (Remarks to the Author): I do not have further comments.

Thank you.